# Contrastive Retrospection: honing in on critical steps for rapid learning and generalization in RL

**Chen Sun**[*]
Mila, Université de Montréal
sunchipsster@gmail.com

**Wannan Yang**
New York University
winnieyangwn96@gmail.com

**Thomas Jiralerspong**
Mila, Université de Montréal
thomas.jiralerspong
@mila.quebec

**Dane Malenfant**
McGill University
dane.malenfant@mila.quebec

**Benjamin Alsbury-Nealy**
University of Toronto, SilicoLabs Incorporated
benjamin.alsbury.nealy@silicolabs.ca

**Yoshua Bengio**
Mila, Université de Montréal, CIFAR
yoshua.bengio@mila.quebec

**Blake Richards**[*]
Mila, McGill University
Learning in Machines & Brains, CIFAR
blake.richards@mila.quebec

## Abstract

In real life, success is often contingent upon multiple critical steps that are distant in time from each other and from the final reward. These critical steps are challenging to identify with traditional reinforcement learning (RL) methods that rely on the Bellman equation for credit assignment. Here, we present a new RL algorithm that uses offline contrastive learning to hone in on these critical steps. This algorithm, which we call Contrastive Retrospection (ConSpec), can be added to any existing RL algorithm. ConSpec learns a set of prototypes for the critical steps in a task by a novel contrastive loss and delivers an intrinsic reward when the current state matches one of the prototypes. The prototypes in ConSpec provide two key benefits for credit assignment: (i) They enable rapid identification of all the critical steps. (ii) They do so in a readily interpretable manner, enabling out-of-distribution generalization when sensory features are altered. Distinct from other contemporary RL approaches to credit assignment, ConSpec takes advantage of the fact that it is easier to retrospectively identify the small set of steps that success is contingent upon (and ignoring other states) than it is to prospectively predict reward at every taken step. ConSpec greatly improves learning in a diverse set of RL tasks. The code is available at the link: https://github.com/sunchipsster1/ConSpec.

## 1 Introduction

In real life, succeeding in a given task often involves multiple critical steps. For example, consider the steps necessary for getting a paper accepted at a conference. One must (i) generate a good idea, (ii) conduct mathematical analyses or experiments, (iii) write a paper, and finally, (iv) respond to reviewers in a satisfactory manner. These are the critical steps necessary for success and skipping any of these steps will lead to failure. Humans are able to learn these specific critical steps even though they are interspersed among many other tasks in daily life that are not directly related to the goal. Humans are also able to generalize knowledge about these steps to a myriad of new projects throughout an academic career that can span different topics, and sometimes, even different fields.

---

[*]Co-corresponding authors

37th Conference on Neural Information Processing Systems (NeurIPS 2023).

Though understudied in RL, tasks involving multiple contingencies separated by long periods of time capture an important capability of natural intelligence and a ubiquitous aspect of real life.

But even though such situations are common in real life, they are highly nontrivial for most RL algorithms to handle. One problem is that the traditional approach for credit assignment in RL, using the Bellman equation (58; 7), will take a lot of time to propagate value estimates back to the earlier critical steps if the length of time between critical steps is sufficiently long (48; 46). Even with the use of strategies to span a temporal gap (*e.g.* TD($\lambda$)), Bellman back-ups do not scale to really long-term credit problems. In light of this, some promising contemporary RL algorithms have proposed mechanisms beyond Bellman backup to alleviate the problem of long term credit assignment (3; 28; 48; 12; 10). But as we show (Figs. 4 and A.16) even these contemporary RL methods are often insufficient to solve simple instantiations of tasks with multiple contingencies, indicating that there are extra difficulties for long-term credit assignment when multiple contingencies are present.

As a remedy to this problem, we introduce Contrastive Retrospection (ConSpec), which can be added to any backbone RL algorithm. Distinct from other approaches to credit assignment, ConSpec takes advantage of the assumption exploited by humans that success is often contingent upon a small set of steps: it is then easier to retrospectively identify that small set of steps than it is to prospectively predict reward at every step taken in the environment. ConSpec learns offline from a memory buffer with a novel contrastive loss that identifies invariances amongst successful episodes, *i.e.*, the family of states corresponding to a successfully achieved critical step. To do this, ConSpec learns a state encoder and a series of prototypes[2] that represent critical steps. When the encoded representation of a state is well aligned with one of the prototypes, the RL agent receives an intrinsic reward (52; 47), thereby steering the policy towards achieving the critical steps.

ConSpec is descended from a classical idea: the identification of "bottleneck states", *i.e.* states that must be visited to obtain reward, and their use as sub-goals (38; 41; 50; 59; 5). However, critical steps in non-toy tasks rarely correspond to specific states of the environment. Instead, there can be a large, potentially even infinite, set of states that correspond to taking a critical step in a task. For example, what are the "states" associated with "conducting an experiment" in research? They cannot be enumerated but a function that detects them can be learned. Ultimately, we thus need RL agents that can learn how to identify critical steps without assuming that they correspond to a specific state or easily enumerable set of states. ConSpec, by harnessing a novel contrastive loss, scalably solves this. Our contributions in this paper are as follows:

- We introduce a scalable algorithm (ConSpec) for rapidly honing in on critical steps. It uses a novel contrastive loss to learn prototypes that identify invariances amongst successful episodes.

- We show that ConSpec greatly improves long-term credit assignment in a wide variety of RL tasks including grid-world, Atari, and 3D environments, as well as tasks where we had not anticipated improvements, including Montezuma's Revenge and continuous control tasks.

- We demonstrate that the invariant nature of the learned prototypes for the critical steps enable zero-shot out-of-distribution generalization in RL.

## 2 Related work

ConSpec is a relatively simple design, but it succeeds because it centralizes several key intuitions shared with other important works. ConSpec shares with bottleneck states (38), hierarchical RL (HRL), and options discovery (59; 5; 41; 50) the idea that learning to hone in on a sparse number of critical steps may be beneficial. But, unlike bottleneck state solutions, ConSpec does not assume that critical steps correspond to individual states (or small, finite sets of states) of the environment. In contrast, ConSpec identifies critical steps in complex, high-dimensional tasks, such as 3D environments. As well, in HRL and options discovery, how to discover appropriate sub-goals at scale remains an unsolved problem. Critical steps discovered by ConSpec could theoretically be candidate sub-goals.

ConSpec shares with Return Decomposition for Delayed Rewards (RUDDER) (3; 44; 62), Temporal Value Transport (TVT) (28), Synthetic Returns (SynthRs) (48), and Decision Transformers (DTs) (12) the use of non-Bellman-based long-term credit assignment. ConSpec shares with slot-like

---

[2]We use the word "prototype" here in the psychological sense, *i.e.* an idealized version of a concept, which in our case, is modelled with a learned vector.

attention-based algorithms such as Recurrent Independent Mechanisms (RIMs) and its derivatives (23; 24) the use of discrete modularization of features. But, unlike all these other contemporary algorithm, ConSpec aims to directly focus on identifying critical steps, a less burdensome task than the typical modelling of value, reward, or actions taken for all encountered states.

ConSpec was inspired, in part, by the contrastive learning literature in computer vision (26; 13; 61; 27). Specifically, it was inspired by the principle that transforming a task into a classification problem and using well-chosen positive and negative examples can be a very powerful means of learning invariant and generalizable representations. A similar insight on the power of classification led to a recent proposal for learning affordances (32), which are arguably components of the environment that are required for critical steps. As such, ConSpec shares a deep connection with this work. Along similar lines, (56; 2; 54; 19; 21; 18; 36; 1) have begun to explore contrastive systems and binary classifiers to do imitation learning and RL. With these works, ConSpec shares the principle that transforming the RL problem into a classification task is beneficial. But these works typically use the contrastive approach to learn whole policies or value functions. In distinction to this, ConSpec uses contrastive learning for the purpose of learning prototype representations for recognizing critical steps in order to shape the reward function, thereby enabling rapid learning and generalization.

## 2.1 Neuroscience inspiration

Our design of ConSpec was ultimately inspired at a high level by theories from neuroscience. For example, it shares with episodic control and related algorithms (37; 35; 8; 45; 40; 51; 43) the principle that a memory bank can be exploited for fast learning. But unlike episodic control, ConSpec exploits memory not for estimating a surrogate value function, but rather, for learning prototypes that hone in on a small number of discrete critical steps. In-line with this, ConSpec takes inspiration from the neuroscience literature on the hippocampus, which suggests that the brain, too, endeavours to encode and organize episodic experience around discrete critical steps (57; 63; 64; 20). Interestingly, recent studies suggest that honing in on discrete critical states for success may in fact be key to how the brain engages in reinforcement learning altogether (29). Contemporary RL is predicated on the Bellman equation and the idea of reward prediction. But recent evidence suggest that dopamine circuits in the brain, like ConSpec, focuses retrospectively on causes of rewards (29), *i.e.* critical steps.

# 3 Description and intuition for ConSpec

Why the use of contrastive learning in ConSpec? Humans are very good, and often very fast, at recognizing when their success is contingent upon multiple past steps. Even children can identify contingencies in a task after only a handful of experiences (55; 22). Intuitively, humans achieve this by engaging in retrospection to examine their past and determine which steps along the way were necessary for success. Such retrospection often seems to involve a comparison: we seem to be good at identifying the differences between situations that led to success and those that led to failure, and then we hone in on these differences. We reasoned that it would be possible for RL agents to learn to identify critical steps—and moreover, to do so robustly—if a system were equipped with a similar, contrastive, capability to take advantage of memory to sort through the events that distinguished successful versus failed episodes. This paper's principal contribution is the introduction of an add-on architecture and loss for retrospection of this sort (Fig. 1).

At its core, ConSpec is a mechanism by which a contrastive loss enables rapid learning of a set of prototypes for recognizing critical steps. Below we describe how ConSpec works.

## 3.1 A set of invariant prototypes for critical steps

To recognize when a new observation corresponds to a critical step, we store a set of $H$ prototypes in a learned representation space. Each prototype, $h_i$ is a learned vector that is compared to a non-linear projection of the latent representation of currently encoded features in the environment, $g_\theta(z_t)$ (with projection parameters $\theta$). Thus, if we have current observation $O_t$, encoded as a latent state $z_t = f_W(O_t)$ (with encoder parameters $W$), we compare $g_\theta(z_t)$ to each of the prototypes, $h_i$, in order to assess whether the current observation corresponds to a critical step or not.

At first glance, one may think that a large number of prototypes is needed, since there are a massive number of states corresponding to critical steps in non-toy tasks. But, by using cosine similarity

between prototype $h_i$ and encoder output $z_t$, $\cos(h_i, g_\theta(z_t))$, ConSpec can recognize a large, potentially infinite set of states as being critical steps, even with a small number of prototypes, because the mapping from $O_t$ to $\cos(h_i, g_\theta(z_t))$ is a many-to-one mapping, akin to a neural net classifier.

In what follows, we used $H \leq 20$ for all our tasks, even the ones in 3D environments and with continuous control, which have a lot of state-level variation. Of course, in other settings more prototypes may need to be used. In general, one can specify more prototypes than necessary, which does not hurt learning (Fig. A.3 for an extreme example).

## 3.2 A memory system for storing successes and failures

ConSpec maintains three memory buffers: one for the current mini-batch being trained on ($\mathcal{B}$; with a capacity of $M_\mathcal{B}$), one for successful episodes ($\mathcal{S}$; with a capacity of $M_\mathcal{S}$), and one for failure episodes ($\mathcal{F}$; with a capacity of $M_\mathcal{F}$). Each episode is of length $T$ time steps. As we show, these memory buffers need not be very large for ConSpec to work; we find that 16 slots in memory is enough, even for 3D environments. Each buffer stores raw observations, $O_t$, encountered by the agent in the environment. When a new mini-batch of observations is loaded into $\mathcal{B}$ for training, the episodes are categorized into successes and failures using the given criteria for the task. We then fill $\mathcal{S}$ and $\mathcal{F}$ with observations from those categorized episodes in a first-in-first-out (FIFO) manner, where episodes stored from previous batches are overwritten as new trajectories come in.

## 3.3 A contrastive loss to learn prototypes

To learn our invariant prototypes of critical steps (and the encoding of the observation) we employ a contrastive loss that differentiates successes from failures. The contrastive loss uses the observations stored in $\mathcal{S}$ and $\mathcal{F}$ to shape the prototypes such that they capture invariant features shared by a spectrum of observations in successful episodes. The functional form of this loss is:

$$\mathcal{L}_{\text{ConSpec}} = \sum_{i=1}^{H} \frac{1}{M_\mathcal{S}} \sum_{k \in \mathcal{S}} |1 - \max_{t \in \{1...T\}} s_{ikt}| + \sum_{i=1}^{H} \frac{1}{M_\mathcal{F}} \sum_{k \in \mathcal{F}} |\max_{t \in \{1...T\}} s_{ikt}| + \\ \alpha \cdot \frac{1}{H} \sum_{k \in \mathcal{S}} D(\{\boldsymbol{s_{ik}}\}_{1 \leq i \leq H}) \tag{1}$$

where $s_{ikt} = \cos(h_i, g_\theta(z_{kt}))$ is the cosine similarity of prototype $h_i$ and the latent representation for the observation from timestep $t$ found in episode $k$, $\boldsymbol{s_{ik}}$ is the vector of $s_{ikt}$ elements concatenated along $t$ and softmaxed over $t$, and $D(\cdot)$ is a function that measures the diversity in the matches to the prototypes (see Appendix section A.5 for functions $D(\cdot)$ we used in this work). Note the purpose of the softmax was to sparsify the sequence of cosine scores for each trajectory over the time-steps $t$. Our goal was to compare pairs of prototypes and force their peak cosine scores to be different in time. The softmax was one way to do this, but any equivalent method would do equally well.

To elaborate on this loss, for each prototype $h_i$ and each episode's projection at time step $t$, $g_\theta(z_{kt})$, the cosine similarity (a normalized dot product) is calculated between the prototype's vector and the latent projection vectors to yield a time-series of cosine similarity scores. The maximum similarity score across time within each episode is then calculated. The maximum similarity score for each successful episode is pushed up (1st term of the loss) and for each failed episode, it is pushed down (2nd term). The 3rd term of the loss encourages the different prototypes to be distinct from one another. The ConSpec loss is added to any RL losses during training (Algorithm 1 pseudocode). Below we show that this loss leads to prototypes that hone in on distinct critical steps (Fig. 4b).

There is a potential nonstationarity in the policy that could cause instability in the prototypes discovered, as the policy improves and previous successes become considered failures. To prevent this, we froze the set of memories that defined each prototype upon reaching the criterion defined in Appendix A.5. In practice, surprisingly, ConSpec does not suffer from learning instability in the multikey-to-door task even if prototype freezing is not done (see Fig. A.12) but it is imaginable that other tasks may still suffer so the freezing mechanism was kept.

## 3.4 Generating intrinsic rewards based on retrospection

Thus far, the described model learns a set of prototypes for critical steps along with an encoder. This system must still be connected to an RL agent that can learn a policy ($\pi_\phi$, parameterized by

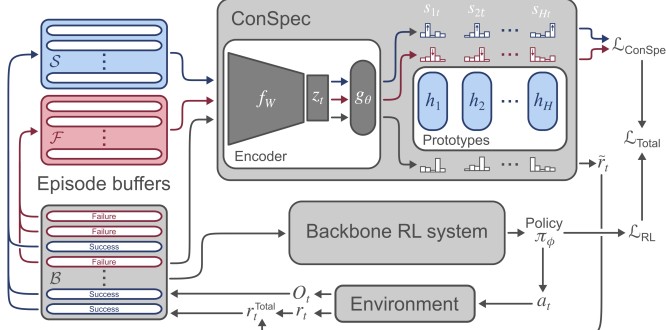

Figure 1: ConSpec is a module that learns to hone in on critical state. ConSpec trains its prototypes by comparing successful v. failed episodes via a contrastive loss, and learning incentivizes pushing cosine similarity scores of successes to 1, and failures to 0. It then uses the match to these prototypes to output intrinsic rewards to the RL agent.

$\phi$). This could proceed in a number of different ways. Here, we chose the reward shaping approach (52; 47; 42), wherein we add additional intrinsic rewards $\tilde{r}_{kt}$ to the rewards from the environment, $r_{kt}$, though other strategies are possible. One simple approach is to define the intrinsic reward in episode $k$ at time-step $t$, $\tilde{r}_{kt}$, as the maximum prototype score, $\hat{s}_{ikt}$, across a moving window of $H$ time-steps ($H = 7$ in our experiments, with a threshold of 0.6 applied to ignore small score fluctuations), and scaled to be in the correct range of the task's expected nonzero reward per step $R_{task}$:

$$\tilde{r}_{kt} = \lambda \cdot R_{task} \sum_{i=1}^{H} \hat{s}_{ikt} \cdot \mathbb{1}_{\hat{s}_{ikt}=\max\{\hat{s}_{ik,t-3},\dots,\hat{s}_{ik,t+3}\}} \tag{2}$$

However, a concern with any reward shaping is that it can alter the underlying optimal policy (47; 42). Therefore, we experimented with another scheme for defining intrinsic rewards:

$$\tilde{r}_{kt} = \lambda \cdot R_{task} \sum_{i=1}^{H} (\gamma \cdot \hat{s}_{ik,t} - \hat{s}_{ik,t-1}) \tag{3}$$

where $\lambda$ is a proportionality constant and $\gamma$ is the discount in RL backbone. This formula satisfies the necessary and sufficient criterion from (42) for policy invariance and provably does not alter the optimal policy of the RL agent. In practice, we find that both forms of intrinsic reward work well (Fig. 4 and A.11) so unless otherwise noted, results in this paper used the simpler implementation, Eqn. 2.

### 3.5 Inductive biases in ConSpec that enable rapid credit assignment

What is the intuition for why ConSpec's inductive biases make it well suited to learn when success is contingent upon multiple critical steps? Learning a full model of the world and its transitions and values is very difficult, especially for a novice agent. Most other RL approaches can be characterized, at a high level, as learning to predict rewards or probability of success given sequences of observations in memory. In contrast, ConSpec solves the reverse problem of retrospectively predicting the critical steps given success. This is a much easier problem, because in many situations, critical steps are independent of one another when conditioned on success. As such, when we consider the joint distribution between success and all of the critical steps, we can learn it retrospectively by considering the conditional probability given success for each potential critical step individually. In contrast, if we learn the joint distribution prospectively, then we must learn the conditional probability of success given all of the potential critical steps in tandem, which is a much more complicated inference problem. To see a concrete example, conditioned on the successful acceptance of a paper, one can be sure that all the critical steps had each individually been achieved (an idea conceived, experiments run, paper written and reviewer concerns addressed). Given that all of the prototypes are learned in parallel, this means that the search for each of these critical steps do not depend on one another when taking this retrospective approach. This is a large gain over any algorithm attempting to learn the function of how combinations of steps could predict reward.

## 4 Empirical results

We tested ConSpec on a variety of RL tasks: grid worlds, continuous control, video games, and 3-D environments. Training was done on an RTX 8000 GPU cluster and hyperparameter choices are detailed in the Appendix A.5. We show how ConSpec rapidly hones in on critical steps, helping alleviate two difficult RL problems: long term credit assignment, and out-of-distribution generalization.

---

**Algorithm 1** Jointly training ConSpec with an RL agent

---

**Given:** Current parameters ($W, \theta, \phi, h_{1...H}$) and **Inputs:** memory buffers $\mathcal{B}, \mathcal{S}$, and $\mathcal{F}$, number of episodes in a mini-batch, $B$, and number of epochs to train for, $E$

1: **for** Epoch $e = 1 \ldots E$ **do**
2:     Collect a new minibatch of $B$ trajectories using $\pi_\phi$ and store them $\{(O_{1:T}, a_{1:T}, r_{1:T})\}$ in $\mathcal{B}$
3:     Update the $\mathcal{S}$ and $\mathcal{F}$ memory banks based on $\mathcal{B}$
4:     Calculate latent representations: $z_{kt} \leftarrow f(O_{kt})$
5:     Calculate scores: $s_{ikt} \leftarrow cos(h_i, g_\theta(z_{kt}))$
6:     Calculate intrinsic rewards, $\tilde{r}_{kt}$ and total rewards, $r_{kt}^{\text{Total}} = r_{kt} + \tilde{r}_{kt}$
7:     Calculate RL loss, $\mathcal{L}_{\text{RL}}$, (per RL backbone) and $\mathcal{L}_{\text{ConSpec}}$ loss per equation 1
8:     Update parameters via loss $\mathcal{L}_{\text{total}} = \mathcal{L}_{\text{RL}} + \beta\mathcal{L}_{\text{ConSpec}}$
9: **end for**

---

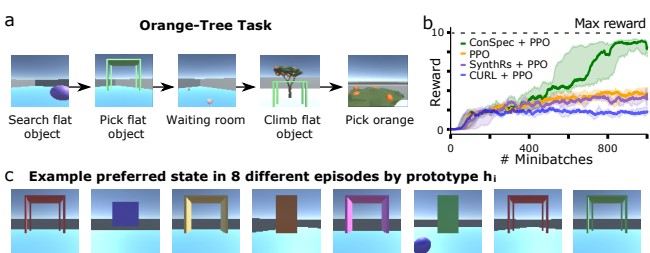

Figure 2: ConSpec learns invariant representations. (a) 3D Orange-Tree task design. (b) ConSpec efficiently learns this task approaching the maximum reward (=10), while PPO, SynthRs, and CURL baselines found it difficult. (c) This prototype learns to robustly detect the retrieval of flat objects, invariant to variations in object type and colour.

## 4.1 ConSpec learns prototypes for critical steps in 3D environments

We began with a series of experiments in a 3D virtual environment. This was to verify that ConSpec can identify critical steps in complex sensory environments, wherein critical steps do not correspond to specific observable states (as bottleneck states do). To do this, we used SilicoLabs'[3] Unity-based (31) game engine to create a 3-stage task that we call the "OrangeTree" task. In stage 1 of this task, the agent is placed in a room with two objects, one of them round or spiky (*e.g.* balls or jacks), and one of them flat (*e.g.* boxes or tables) (Fig. 2a) and the agent can pick one of them up. In stage 2, the agent is put in a waiting room where nothing happens. In stage 3, the agent is placed in a new room with an orange tree. To obtain a reward, the agent must pick an orange, but to do this, it must stand on a flat object (which it can do only if it had picked up a flat object in stage 1). So in stage 1, the agent must learn the critical step of picking up a flat object. But, because there are a variety of flat objects, there is no single bottleneck state to learn, but rather, a diverse set of states sharing an invariant property (*i.e.* object flatness).

ConSpec was compared with a series of baselines. We first tested Proximal Policy Optimization (PPO) (53; 33) on this task, which failed (Fig. 2b). One possibility for this is the challenge of long-term credit assignment, which arises from the use of the waiting room in stage 2 of the task. As such, we next tested an agent trained with Synthetic Returns (SynthRs) (48) atop PPO, a recent reward shaping algorithm designed to solve long-term credit assignment problems. SynthRs works by learning to predict rewards based on past memories and delivering an intrinsic reward whenever the current state is predictive of reward. But SynthRs did not help solve the problem (Fig. 2b), showing that the capacity to do long-term credit assignment is, by itself, not enough when success does not depend on a single bottleneck state but rather a diverse set of states that correspond to a critical step.

Thus, we next considered more sophisticated representation learning for RL: Contrastive Unsupervised Reinforcement Learning (CURL) (56), which outperforms prior pixel-based methods on complex tasks. However, adding CURL to PPO, too, did not solve the task (Fig. 2b), likely because CURL is not learning invariances for task success, but rather, invariances for image perturbations.

Finally, we examined the performance of ConSpec. An agent trained by PPO with additional intrinsic rewards from a ConSpec module (per Algorithm 1) can indeed solve the OrangeTree task (Fig. 2b). To further dissect the reasons for ConSpec's abilities, we studied the invariances it learned. To do this, we collected successful episodes and then identified the range of observations (images) that maximally matched the prototypes during the episodes. Notably, one of the prototypes consistently

---

[3]https://www.silicolabs.ca/

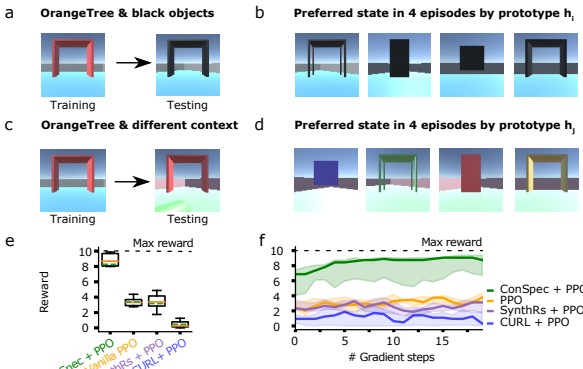

Figure 3: ConSpec's invariant representations help OoD generalization. (a, c) New OrangeTree tasks during Testing (a) with black boxes/tables, or (c) in a differently coloured room. Previously trained prototypes hone in on interpretable states even in these new contingencies (b,d) and are able to generalize (e) in zero-shot with never-before seen black objects, and (f) in few-shot in a new environment, approaching the max reward.

had observations of picking up a flat object mapped to it. In other words, this prototype had discovered an invariant feature of a critical step that was common to all successes but not failures, namely, the step of having picked up a flat object (Fig. 2c). Moreover, this prototype learned to prefer flat objects regardless of the type of object (*i.e.* box or table) and regardless of its colour. So, ConSpec's contrastive learning permitted the prototype to learn to recognize a critical step, namely picking up flat objects, and to maintain invariance to irrelevant features of those objects such as shape and colour.

## 4.2   ConSpec generalizes to new contingencies in 3D environments

We next asked whether ConSpec's capacity for finding invariances among critical steps could aid in zero-shot generalization when the sensory features of the environment were altered. To test this, we made a variation of the OrangeTree task. During training, the agent saw flat objects that were magenta, blue, red, green, yellow, or peach (Fig. 3a). During testing, a black table or box was presented, even though the agent had never seen black objects before. The prototype that recognized picking up flat objects immediately generalized to black flat objects (Fig. 3b). Thanks to this, the agent with ConSpec solved the task with the black objects in zero-shot (Fig. 3e). This shows that ConSpec was able to learn prototypes that discovered an invariant feature among successes (flatness), and was not confused by irrelevant sensory features such as colour, permitting it to generalize to contingencies involving colours never seen before.

To further test ConSpec's capacity for generalization, we made another variation of the OrangeTree task with the background environment changed, in a manner typical of out-of-distribution (OoD) generalization tests (4). In particular, testing took place in a different room than training, such that the new room had pink walls and a green floor (unlike the gray walls and blue floor of the training room) (Fig. 3c). Again, the prototypes generalized in a zero-shot manner, with flat objects being mapped to the appropriate prototype in the new environment despite the different background (Fig. 3d). The policy network did not immediately generalize, but after a brief period of acclimatization to the new environment (where the underlying PPO agent was trained for no more than 20 gradient descent steps while ConSpec was kept constant), the ConSpec agent was able to rapidly solve the new but related task environment (Fig. 3f).

## 4.3   ConSpec improves credit assignment in gridworld tasks with multiple critical steps

To investigate more deeply ConSpec's ability to engage in long-term credit assignment by identifying critical steps, we used simple multi-key-to-door tasks in 2D grid-worlds. These tasks were meant to capture in the simplest possible instantiation the idea that in real life, rewards are sparse and often dependent on multiple critical steps separated in time (akin to the paper-acceptance task structure described in the Introduction 1). In the multi-key-to-door tasks we used here, the agent must find one or more keys to open one or more doors (Fig. 4a). If the agent does not pick up all of the keys and does not pass through all of the doors, success is impossible. To make the tasks challenging for Bellman back-up, the agent is forced to wait in a wait room for many time steps in-between the retrieval of each key. An episode is successful if and only if the agent exits the final door.

We found that when ConSpec was trained in these tasks, each prototype learned to represent a distinct critical key-retrieval step (*e.g.* one prototype learned to hone in on the retrieval of the first key, another prototype on the second key, etc.). We demonstrated this by examining which state in different episodes maximally matched each prototype. As can be seen, picking up the first key was

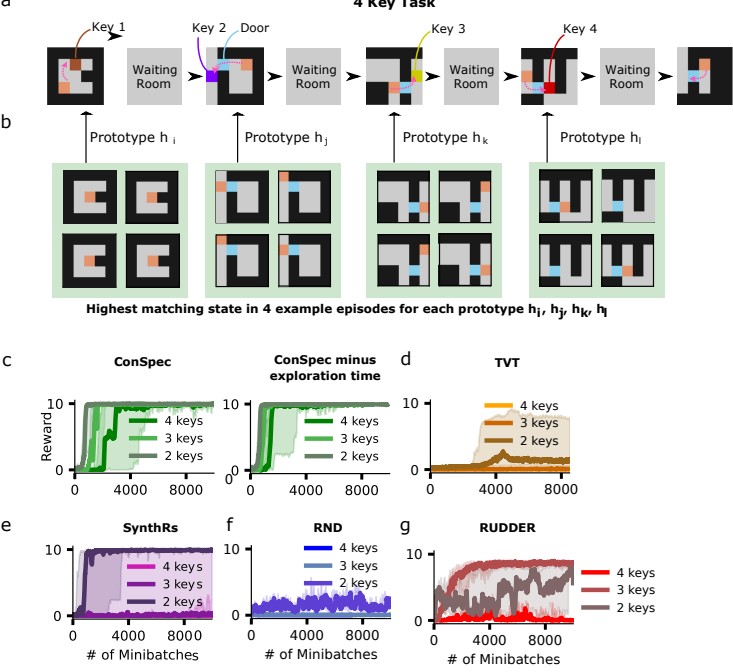

Figure 4: ConSpec rapidly learns tasks with multiple contingencies akin to the scenario from the Introduction 1 (a) Protocol for multi-key-to-door task with 4 keys. (b) Plotted: states that maximally matched each prototype. Here, prototypes learn to hone in on states that depict retrieval of the key (where the agent is out the door and the coloured keys are gone *i.e.* retrieved). (c *left*) ConSpec rapidly learns the multi-key-to-door tasks, whereas (d-g) TVT, SynthRs, RND, and RUDDER performances collapse. (c *right*) When exploration time is subtracted, ConSpec's training time $\propto \mathcal{O}(constant)$ approximately, even as number of keys increased, affirming the complexity predictions from 3.5.

matched to one prototype, picking up the second was matched to another, and so on (Fig. 4b). Thus, ConSpec identifies multiple critical steps in a highly interpretable manner in these tasks.

We then tested ConSpec atop PPO again. We compared performance to four other baselines. The first three, SynthRs, Temporal Value Transport (TVT) system (28), and RUDDER (3), are contemporary reward-shaping based solutions to long-term credit assignment in RL. The fourth baseline was random network distillation (RND) (10), a powerful exploration algorithm for sparse reward tasks, in order to study if exploration itself was sufficient to overcome credit assignment in situations with multiple critical steps.

We found that PPO with ConSpec learns the multi-key-to-door tasks very rapidly, across a range of number of keys (Fig. 4c, *left*). Importantly, even as keys got added, ConSpec's training time remained constant when the exploration required for new keys was accounted for (Fig. 4c, *right*), consistent with the intuitions given above (section 3.5). On the other hand, RUDDER, SynthRs, TVT, and RND solved the task for a single key thanks to their ability to do long-term credit assignment (Fig. A.2, A.4, A.5), but their performance collapsed as more keys were added (Fig. 4d-g), highlighting the difficulty of long-term credit assignment with multiple critical steps.

We also note that RND, despite being strong enough to handle sparse reward tasks like Montezuma's Revenge (10), also fails here as more keys are added, illustrating the point that long term credit assignment is its own distinct issue beyond exploration which ConSpec addresses and RND does not.

What happens when the number of prototypes is not sufficient to cover all the critical steps? Even having fewer than necessary prototypes (3 prototypes in the 4-key task, shown in Fig. A.8) can surprisingly still be enough to solve the task (i.e. catching any critical step at all, still helps the agent).

We also tested ConSpec on another set of grid-world tasks with multiple contingencies, but with a different and harder to learn task structure. Again, ConSpec but not other baselines could solve these tasks (Fig. A.16), illustrating that ConSpec successfully handles a wide variety of real-life inspired scenarios with multiple contingencies.

Finally, we note that ConSpec successfully learns, not only atop a PPO backbone, but also atop an Reconstructive Memory Agent (RMA) backbone (Fig. A.10 and also Fig. A.2b), illustrating ConSpec's flexibility in being added to any RL agent to improve long-term credit assignment.

### 4.4   ConSpec helps credit assignment in delayed Atari tasks and Montezuma's Revenge

Next we studied how ConSpec could help credit assignment in other RL settings, such as the Atari games from OpenAI Gym (9), including in some unanticipated settings where it would not have been expected to help. We first note that existing RL algorithms, such as PPO, can solve many of these games, as does PPO with ConSpec (Fig. A.17). But, PPO alone cannot solve delayed reward versions of the Atari games. These require specialized long-term credit assignment algorithms to solve them (3; 48). And adding ConSpec to PPO also solves delayed Atari (Fig. 5).

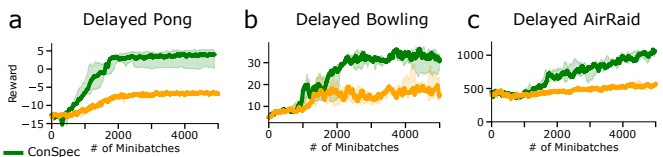

Figure 5: ConSpec improves performance on delayed (a) Pong, (b) Bowling, and (c) AirRaid.

Within the set of unmodified Atari games, PPO alone also does not solve Montezuma's Revenge, and it does not usually exceed 0 reward, so we investigated this game further. Montezuma's Revenge is usually considered to be an exploration challenge, and sophisticated exploration algorithms have been developed that solve it like RND(10) and Go-Explore (16). But, interestingly, we see that adding ConSpec to PPO improves performance on Montezuma's Revenge even without any sophisticated exploration techniques added (Fig. 6a). To provide intuition as to why ConSpec is able to help even in the absence of special exploration, we studied how many successful trajectories it takes for ConSpec to learn how to obtain at least the first reward in the game. Crucially, ConSpec atop PPO with only a simple $\epsilon$-greedy exploration system is able to learn with as few as 2 successful episodes (Fig. 6b). We speculate that this is because ConSpec, like other contrastive techniques (13), benefits mostly from a large number of *negative* samples (*i.e.* failures) which are easy to obtain. Other algorithms like Recurrent Independent Mechanisms (RIMs) have demonstrated large improvements on other Atari games (23), and make use of discrete modularization of features as ConSpec does by its prototypes. But RIMs discretizes by a very different mechanism, and does not make progress on Montezuma's Revenge (Fig. 6c).

Interestingly, Decision Transformers (DT) are designed to learn from successful demonstrations as ConSpec does (Fig. 6d) but cannot solve Montezuma's Revenge if those successes were "spurious successes" (an important learning signal for novice agents) during random policies, affirming that DTs, unlike ConSpec, need to be behaviourally cloned from curated experts. This was still true when the DTs were trained with every success trajectory uniformly labelled +1 and every failure 0. Even with this equally weighting of data, DTs still got 0 reward. (Fig. 6e) By contrast, ConSpec learns well by inherently **treating the data unequally**, honing in on only a few critical steps.

Putting this altogether, the secret to ConSpec's rapid abilities to learn from spurious successes arises because the prototypes ignore irrelevant states and learn to hone in on the few critical ones, allowing ConSpecto get reward on challenging tasks like Montezuma's Revenge, doing so even without a dedicated exploration algorithm. We speculate that coupled with an exploration algorithm ConSpec could be even more powerful.

### 4.5   ConSpec helps credit assignment in continuous control tasks

Finally, we study another scenario in which ConSpec unexpectedly helps. Although ConSpec was designed to exploit situations with discrete critical steps, we studied whether ConSpec might help even in tasks that do not explicitly involve discrete steps, such as continuous control tasks. These tasks are interesting because both the robotics and neurophysiology literature has long known that "atomic moves" can be extracted from the structure of continuous control policies (15; 60). Moreover, this is evident in real life. For example, learning to play the piano is a continuous control task, but we can still identify specific critical steps that are required (such as playing arpeggios smoothly). However, contemporary RL algorithms for continual control tasks are not designed to hone in on sparse critical steps, and instead, they focus on learning the Markovian state-to-state policy.

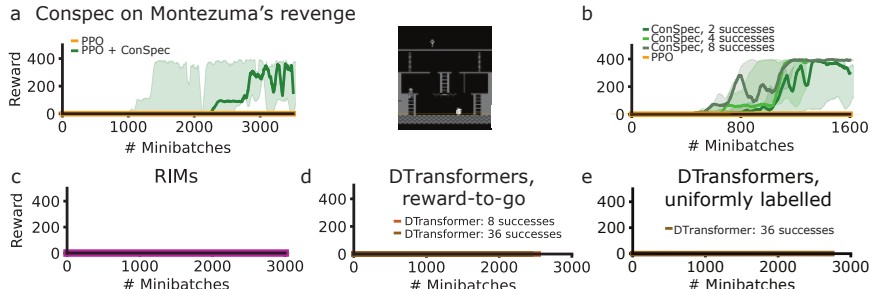

Figure 6: (a) PPO+ConSpec progresses on Montezuma's Revenge despite the lack of a dedicated exploration algorithm. (b) PPO+ConSpec with 2, 4, and 8 success demonstrations, highlighting how ConSpec uses few successes very efficiently. (c) RIMs, and (d) DTs get 0 reward, even when DTs are given up to 36 spurious success demonstrations. (e) DTs trained with success trajectories uniformly labelled +1 and failures 0. With this uniform weighting of the data, DTs still get 0 reward.

We focused on delayed versions of three continuous control tasks: Half Cheetah, Walker, and Hopper, where the delay was implemented according to past literature (49) to introduce a long-term credit challenge (see A.4). ConSpec atop PPO is able to significantly improve performance in all tasks (Fig. 7), showing the utility of learning to identify critical steps even in continuous control tasks.

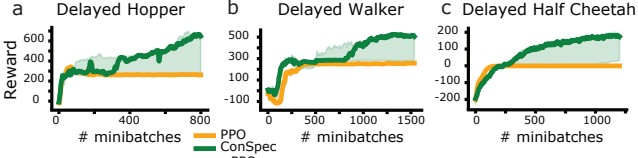

Figure 7: ConSpec improves performance in delayed versions of continuous control tasks (a) Hopper, (b) Walker, (c) Half Cheetah.

# 5 Discussion and limitations

Here, we introduce ConSpec, a contrastive learning system that can be added to RL agents to help hone in on critical steps from sparse rewards. ConSpec works by learning prototypes of critical steps with a novel contrastive loss that differentiates successful episodes from failed episodes. This helps both rapid long-term credit assignment and generalization, as we show. Altogether, ConSpec is a powerful general add-on to any RL agent, allowing them to solve tasks they are otherwise incapable.

Despite its benefits for rapid credit assignment and generalization beyond Bellman-backup-based approaches in RL, ConSpec has limitations, which we leave for future research. For one, the number of prototypes is an unspecified hyperparameter. In our experiments, we did not use more than 20, even for rich 3D environments, and in general one can specify more prototypes than necessary, which does not hurt learning (Fig. A.3) for an extreme case). However, an interesting question is whether it would be possible to design a system where the number of prototypes can change as needed. For another limitation, ConSpec requires a definition of success and failure. We used relatively simple definitions for tasks with sparse rewards as well as for tasks with dense rewards as detailed in section A.3, and they worked well. But, future work can determine more principled definitions. Alternatively, definitions of success can be learned or conditioned, and as such, are related to the topic of goal-conditioned RL, which is left unexplored here. Nonetheless, we believe that this is a promising future direction for ConSpec, and that it may potentially be of great help in the yet unsolved problem of sub-goal and options discovery.

# 6 Acknowledgements

We heartily thank Doina Precup, Tim Lillicrap, Anirudh Goyal, Emmanuel Bengio, Kolya Malkin, Moksh Jain, Shahab Bakhtiari, Nishanth Anand, Olivier Codol, David Yu-Tung Hui, as well as members of the Richards and Bengio labs for generous feedback. We acknowledge funding from the Banting Postdoctoral Fellowship, NSERC, CIFAR (Canada AI Chair and Learning in Machines & Brains Fellowship), Samsung, IBM, and Microsoft. This research was enabled in part by support provided by (Calcul Québec) and the Digital Research Alliance of Canada.

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

# A   Appendix

## A.1   Reproducibility

We aim to maximize the reproducibility of this study. A detailed description and full pseudo-code of ConSpec is given in the method section and appendix. The ConSpec code is available at the link: https://github.com/sunchipsster1/ConSpec.

## A.2   Broader Impact

The focus of this work is the introduction of a new add-on algorithm in reinforcement learning to help better generalization and long term credit assignment. We believe that there is potential for positive impact as it can help make many real-life problems, which involve long-term credit assignment, more solvable. However, it could be conceivably applied to ethically-questionable problems in RL. Users of such methods must be aware when applying this method to their scientific problems.

## A.3   Defining success and failure

How best to define successful and unsuccessful episodes? There are three fairly obvious ways:

1. In the domain of sparse reward tasks, the definition of a "successful" episode is relatively straightforwardly defined as an episode that received one of the few rewards available, and a "failure" episode is defined as anything else.

2. In non-sparse reward settings, the notion of a "successful" episode can be defined as the top set of $k$ episodes based on the sum of rewards. As such, the trajectories defined as successful are updated as an agent improves. Therefore, ConSpec uses the reward signal in a comparative and relativistic way, giving it the ability to hone in on key steps associated with the best performance achieved so far.

3. In a goal-conditioned setting, a "successful" episode is simply one which achieves the goal.

In ConSpec  we use the first approach, since the focus of this work is, in part, on solving tasks with sparse rewards and multiple contingencies leading to them. In two cases we do take the second approach (with Atari games and Continuous control tasks) (9) to demonstrate that this can also work. We have yet to explore the third approach, which we leave for future research.

## A.4   Task implementation details

OrangeTree experiments were implemented using SilicoLabs' Unity ((31)) based game engine to create virtual 3D environments. In stage 1 of this task, the agent sees two objects around the room, one of them round or spiky, and one of them flat (*e.g.* boxes or tables), and the agent can pick up one of the objects (Fig. 2). In stage 2, the agent is again put in a waiting room. In stage 3, the agent must obtain oranges from the tree, but to do so, it must stand on a flat object in order to reach the oranges and receive reward. Successful completion of the task resulted in a reward of 10. Observations given to the RL agent were 84x84x3 pixels. The tasks were a total of 65 timesteps long, with 15 timesteps in Stage 1, 40 timesteps in Stage 2, and 10 timesteps in Stage 3.

In the multi-key-to-door tasks the agent must find all the keys to open the doors (Fig. 4a). If the agent does not pick up all of the keys and pass through all of the doors, success is impossible. In between each stage for picking up keys/passing through doors, the agent is forced to wait in a waiting room for many time steps in between the retrieval of each key. An episode is successful if and only if the agent exits the final door. Observations given to the RL agent were minimalist, at 5x5x3 pixels. Successful completion of the task resulted in a reward of 10. Tasks had alternating key retrieval and waiting periods, and the lengths of each of these stages are indicated here A.1.

Atari tasks were implemented with frameskips of 4, and for computational expediency, and were uniformly terminated at 600 timesteps at which point rewards were tallied. Observations given to the RL agent were 84x84x3 pixels (where the 3 encodes colours RGB). For Delayed Atari games, episodes were terminated after 600 timesteps and cumulative rewards were given only at the end, similar to previous works (3; 48).

Number of timesteps for each task stage:

| Stage | Baseline key-to-door task | 2-key task | 3-key task | 4-key task |
|---|---|---|---|---|
| Retrieve 1st key | 10 | 10 | 10 | 10 |
| Waiting period | 85 | 85 | 85 | 85 |
| Retrieve 2nd key | | 15 | 15 | 15 |
| Waiting period | | 85 | 85 | 85 |
| Retrieve 3rd key | | | 15 | 15 |
| Waiting period | | | 85 | 85 |
| Retrieve 4th key | | | | 15 |
| Waiting period | | | | 85 |
| Exit the final door | 10 | 10 | 10 | 10 |
| | | | | |
| Total: | 105 | 205 | 305 | 405 |

Figure A.1: Protocols for multi-key-to-door experiments.

Continuous control tasks were run for 250 timesteps. The traditional version of these tasks gives the agent a forward reward at each timestep based on how much it has moved in the forward direction during that timestep. The delayed versions of the tasks– taken from previous literature ((49))– were created by modifying the reward function so that the forward reward is not given at each timestep, but only once the agent has successfully moved a certain number of units in the forward direction (called the threshold). On the timestep that the agent passes a threshold, it receives the cumulative reward that it would have received for running up to that threshold during all the previous timesteps. Since a fully trained agent on Half Cheetah runs approximately three times farther than a fully trained agent hops in Hopper and walks in Walker, the thresholds for Hopper and Walker were set at 1, while the threshold for Half Cheetah was set at 3.

### A.5 Experimental and architectural details

Training on all experiments was done on an RTX 8000 GPU cluster. All seeds were run with 32GB of memory until completion.

$\mathcal{S}$ **and** $\mathcal{F}$ **buffer sizes**: All experiments used $\mathcal{S}$ and $\mathcal{F}$ buffer sizes of 16. For 3D OrangeTree experiments, although $\mathcal{S}$ and $\mathcal{F}$ buffers each held 16 episodes like all the other experiments, each gradient step was done on a random minibatch of 8 episodes sampled from this 16 due to computational resource limitations. In Atari tasks each gradient step was done on a random minibatch of 4 episodes sampled from the buffers of 16 for similar reasons.

**Minibatch size**: all models used standard minibatch size $B = 16$, except for the Atari tasks, which due to computational resource limitations, used minibatch size $B = 8$.

**Seeds in experiments**: All multi-door-to-key experiments averaged over 10 seeds, as did all 3D OrangeTree experiments. The 9 Atari experiments and 3 Continuous control experiments averaged over 5 seeds each. All plots show median and quartile range.

**Number of prototypes**: Across implementations of ConSpec for various tasks unless otherwise indicated, the number of prototypes used was set as 8, except for ConSpec on the task in Fig. A.16 and the implementation of ConSpec on an RMA backbone (Fig A.10), both of which used 16, and the Atari tasks, which due to computational resource limitations, used 3 prototypes.

We also tested ConSpec in the 4-key task with various numbers of prototypes and find that performance was robust across values (Fig. A.7).

**Architectural details for models used**: For all multi-key-to-door tasks, the learned encoder of the underlying RL agent was a single layer 2-D convolutional neural network (outchannels = 32, kernel = 3) with ReLU activation, followed by a final linear layer, and a 512-unit GRU, as was done in the (28) codebase for 2D key-to-door tasks. The convnet encoder for the ConSpec module was

identically sized, but separately parameterized so that there would be no interference between the ConSpec module and the underlying RL agent. In the ConSpec module, the nonlinear projection $g_\theta$ in ConSpec was a 2-layer MLP with 1010 units in the intermediate layers and the final output layer, and ReLU activations between layers. Inspired by the contrastive learning literature, where the importance of having a large number of negative samples relative to positive samples has been observed (13), the success buffers were filled at a slower rate than the failure buffers: at most 2 new successful trajectories were added to $\mathcal{S}$ from each minibatch, while there was no limit to the number of new failed trajectories added to $\mathcal{F}$ from each minibatch.

Continuous control tasks have small observation spaces and therefore did not require a convnet encoder. In continuous control experiments, agents simply used a two-layer 64-unit MLP with tanh nonlinearity.

Atari tasks and 3D gridworld tasks possess larger observations, so the learned encoder used was a much larger Impala encoder (17; 14) using 3 Impala layers with 16, 32, and 32 channels respectively. For these experiments, the nonlinear projection $g_\theta$ in ConSpec was a 2-layer MLP with 100 units in the intermediate layers and the final output, and ReLU activations between layers.

One of the aims of the Atari experiments was to apply ConSpec in a setting that requires a different definition of successes and failures, as elaborated in section A.3. Specifically, successful trajectories were defined as the highest cumulatively scoring ones from each minibatch, while comparatively, failed trajectories were randomly chosen from the remainder of the minibatch. To extend ConSpec to this setup, each of ConSpec's prototypes took not just the $s_{ikt}$ with the top cosine similarity for each trajectory, but rather, averaged the $s_{ikt}$'s with the top-$k$ cosine similarities for each trajectory (where $k = 20$).

For the hyperparameter $\lambda$, which scales the intrinsic reward (see equation 3), we used a different value in each set of experiments (multi-key-to-door, Atari, and OrangeTree). Specifically, we used $\lambda = 0.2$ for the multi-key-to-door, $\lambda = 0.5$ for the policy-invariant version of ConSpec in the multi-key-to-door experiment, $\lambda = 0.5$ for the Atari games, $\lambda = 0.5$ for OrangeTree, and $\lambda = .2$ for Continuous control tasks. These values were determined via a logarithmic grid-search that selected for best reward. Other hyperparameters were found via a logarithmic grid-search, and were shared across all tasks:

- RL agent learning rate: $2 \times 10^{-4}$
- ConSpec learning rate: $2 \times 10^{-3}$
- $\alpha$: 0.2
- $\beta$: 1.0

We also show for the OrangeTree task, the 4-key task, and Montezuma's Revenge task, that in each of these tasks, performance was relatively insensitive to lambda between 0.2 and 0.5 (Fig. A.9).

Models using a PPO backbone used the Adam optimizer with optimal PPO hyperparameters based on the repository (33) (values below). The only major change to the hyperparameters used was the entropy coefficient used (0.02, rather than 0.01 from the repository, in order to encourage exploration necessary in the multi-key-to-door tasks):

- learning rate: $2 \times 10^{-4}$
- Adam epsilon: $1 \times 10^{-5}$
- reward discount factor: 0.99
- clipping 0.08
- value loss coefficient: 0.5
- entropy coefficient: 0.02

PPO-backbone models that were engaged with either the multi-door-to-key tasks or the 3D Orange-Tree task had reward normalized to the range $[0, 1]$. PPO-backbone models that were engaged with the 9 Atari tasks underwent reward clipping to the sign of the reward received at each timestep, as is standard practice (39).

Models using an RMA backbone (including the implementation of TVT) were taken directly from the (28) codebase without modification of the architecture. They used the Adam optimizer with PPO hyperparameters based on the repository (28). The only major change to the hyperparameters used was the entropy coefficient used (0.1, rather than 0.05 from the repository, in order to encourage exploration necessary in the multi-key-to-door tasks):

- learning rate: $2 \times 10^{-4}$
- Adam epsilon: $1 \times 10^{-6}$
- agent discount: 0.92
- clipping 0.1
- reading strength coefficient: 50.
- image strength coefficient: 50.
- entropy coefficient: 0.1

The implementation for SynthRs used a synthetics returns MLP that was sized analogously as $g_\theta$ in ConSpec in the respective experiments (2-layer MLP with ReLU and 1010 or 100 units respectively). This module was put on top of a PPO backbone. Its convolutional neural network encoder was identical to the corresponding encoders used in ConSpec. The version of SynthRs from (48) that did not make use of a bias term $b(s_t)$ was used since in our tasks the reward is totally predictive from the current state (*e.g.* whether the agent exits the final door or not), negating the need for linear regression if the bias term is present. Here, SynthRs learning rates were matched to the learning rates used in ConSpec to enable comparable rates of learning ($2 \times 10^{-4}$ for the underlying PPO agent, and $2 \times 10^{-3}$ for the synthetics return module). The optimal weight of the SynthRs loss, $\beta$, as defined in Algorithm 1 was determined via further logarithmic grid-search, and $\beta = 10^{-4}$ worked well across experiments. Further hyperparameters were as follows, taken amongst the optimal values used in the SynthRs paper (48):

- state-associate alpha: 0.01
- state-associate beta: 1.0

SynthRs was unable to solve either the 3D OrangeTree task nor the mult-key-to-door tasks in the allotted time (Figure A.15a-b). But replacing the sigmoid with a softmax on the last layer of the gate $g$ improved performance on both tasks (Figure 4f and 2b), so this was the version of SynthRs used in all experiments.

The implementation of CURL was taken from the repository (34). We replaced the encoder with an Impala neural network identical to the encoder used by PPO and ConSpec in the 3D OrangeTree experiment. We also replace the SAC agent in the original repo with a PPO agent. Hyperparameters were as follows:

- pretransform image size: 96
- image size: 84
- frame stack: 1
- batch size for curl: 128
- curl latent dim: 128
- encoder tau: 0.05
- framestack: 100000

The implementation of Decision Transformers for the grid-world task was taken from the repository (6), but we added a convolutional neural network encoder that was identical to the encoder used in the ConSpec experiment. Moreover, its learning rate ($2 \times 10^{-4}$) was matched to the learning rate used in ConSpec to enable comparable rates of learning. All other hyperparmeters were taken without modification from the repository (6) and were:

- weight decay: $1 \times 10^{-4}$

- dropout probability: 0.1
- context length: 20
- number of blocks: 3

The implementation of Random Network Distillation was taken from (30) with the standard convolutional neural network encoder that was identical to the encoder used in the ConSpec experiment, All other hyperparameters were the default ones:

- extrinsic reward clipping: $[-1, 1]$
- intrinsic reward clipping: False
- observation clipping after normalization: $[-5, 5]$

The implementation of Decision Transformers for the Atari tasks was taken from (11) with default parameters, which had been already optimized by the authors for Atari games.

The implementation of RIMs for the Atari tasks was taken from (25) with default parameters, which had been already optimized by the authors for Atari games.

**Function $D$ for encouraging diversity**: The function $D$ is used in the loss function, equation 1, for encouraging diversity amongst the prototypes. With the exception of Montezuma's Revenge (which will be discuss separately, below), the function $D$ that was used for all other experiments was $D(\{s_{ik}\}_{1 \leq i \leq H}) = \sum_{i \neq j} \cos(s_{ik}, s_{jk})$ where $\cos(\cdot, \cdot)$ is cosine similarity. This term is therefore an orthogonality constraint imposed to encourage the different prototypes to be independent from each other.

**Prototype freezing: keeping prototypes stable once they are learned**:

There is a potential source of instability in learning prototypes with ConSpec. Specifically, once an agent has learned a prototype for a specific critical step and helped shape the policy so that the agent consistently takes this step, the failure buffer will begin to have many examples where this critical step if it is not by itself sufficient to achieve success (e.g. if multiple keys are required). Thereafter the initial prototype would begin to diverge from this critical step, and this can lead to instability. To prevent this, updates to the memory buffers for each prototype were terminated upon reaching criterion that this prototype sufficiently differentiated between successes and failures above threshold $\mathcal{T}$. Across all experiments, this criterion was defined as when the average maximum cosine similarity scores among trajectories in $\mathcal{S}$- $\mathcal{F}$ differed by $> \mathcal{T}$ and that the scores in $\mathcal{S}$ were themselves $> \mathcal{T}$, all for at least 25 consecutive gradient steps. We found that $\mathcal{T} = 0.6$ worked well in practice, and this value was used through *all* experiments.

This design is neuroscience-motivated because semantic knowledge is often defined by exemplars. The frozen set of episodes for prototype $i$ serve as the examplars that define prototype $i$. In an experiment, we ablated this prototype freezing mechanism, and interestingly found that even without prototype freezing, ConSpec performs well in the presence of multiple contingencies (Fig. A.12).

**Recruitment of prototypes**:

In most experiments, all prototypes were made immediately available to the agent as the simplest case. However, we also tested recruiting prototypes only as needed in the Montezuma's Revenge and Continuous control experiments. In these experiments, only 3 prototypes were made immediately available to train on. Only when the existing prototypes differentiated success from failure above threshold $\mathcal{T}$, would a new prototype be recruited to be newly trained. Therefore, this was a scheme by which the prototypes would only be recruited as needed.

**Montezuma's Revenge**: The purpose of the Montezuma's Revenge experiments was to study whether ConSpec can be used to help Montezuma's Revenge, even when it is put atop PPO, despite the lack of any further dedicated exploration algorithm. Montezuma's Revenge required a greater task time (necessary to encourage as much exploration as possible in the environment) and was terminated at a longer 1200 timesteps. To study this game, the learned encoder was an Impala encoder (17; 14) using 3 Impala layers with 16, 32, and 32 channels respectively, and a 256-unit GRU. For these experiments, the nonlinear projection $g_\theta$ in ConSpec was a 2-layer MLP with 100 units in the intermediate layers and the final output, and ReLU activations between layers. The model used 20 prototypes and $\lambda = 0.2$. The function $D$ that was used was $D(\{s_{ik}\}_{1 \leq i \leq H}) = -H(L_1(\sum_i s_{ik}))$ where $H(\cdot)$ is entropy and

$L_1(\cdot)$ is $L_1$ normalization. This term therefore encourages the different prototypes to hone in on the trajectory as uniformly from each other as possible, thereby encouraging prototype diversity. Successes in Montezuma's Revenge are especially sparse especially to the novice agent, which means that recurrent states stored in the $\mathcal{S}$ buffer are rarely updated (especially initially). To alleviate this, we simply replayed a randomly chosen success trajectory to the agent every training iteration, to enable it to recalculate its stored recurrent states in $\mathcal{S}$ buffer. The training of the ConSpec + PPO model did not initiate until at least 2 successes had been spuriously experienced. The minibatch size used in these experiments was $B = 5$ due to limitations in computational resources, and memory freezing was not done as this would have added more memory buffers due to limitations in computational resources. The Montezuma's Revenge experiment Fig. 6a) used 10 seeds and excluded 4 other seeds that were prematurely terminated by the Pytorch autograd during the learning of the task.

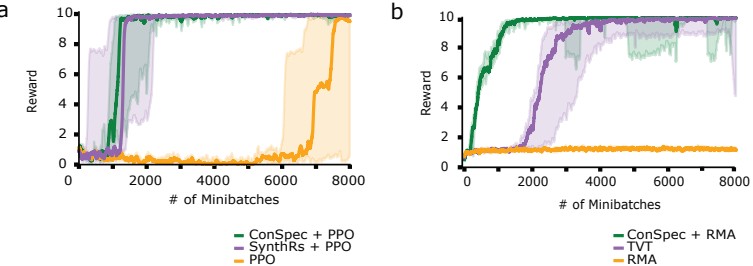

Figure A.2: Baseline single key-to-door task. (a) Performance of ConSpec on a PPO backbone vs SynthRs on a PPO backbone vs PPO. (b) Performance of ConSpec on an RMA backbone vs TVT vs RMA.

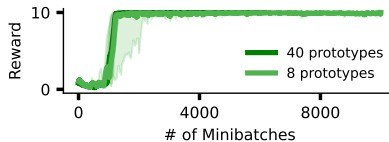

Figure A.3: Performance of ConSpec when the number of prototypes is varied (40 vs 8) does not change, in the vanilla single key-to-door task.

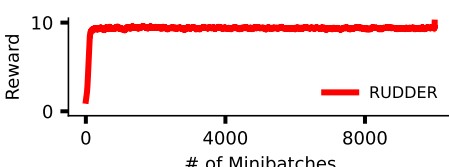

Figure A.4: Performance of RUDDER on the baseline single key-to-door task.

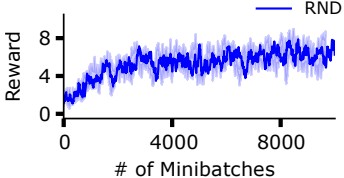

Figure A.5: Performance of RND on the baseline single key-to-door task.

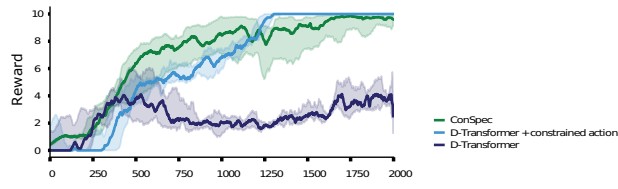

Figure A.6: ConSpec is more robust than Decision Transformers on vanilla key-to-door because of its objective in learning states rather than predicting actions. Decision transformers is not able to solve the key-to-door task unlike ConSpec but is able to solve it if actions are constrained in their freedom.

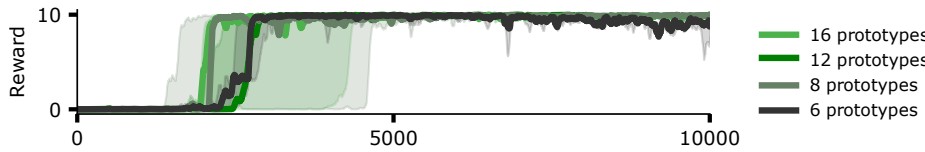

Figure A.7: ConSpec performance was relatively insensitive to changes in the number of prototypes in the 4-key task.

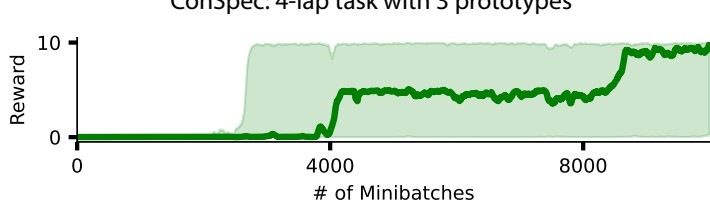

Figure A.8: ConSpec's performance on the 4-key task with only 3 prototypes: even having fewer than necessary prototypes can surprisingly often catch enough critical steps to still solve the task.

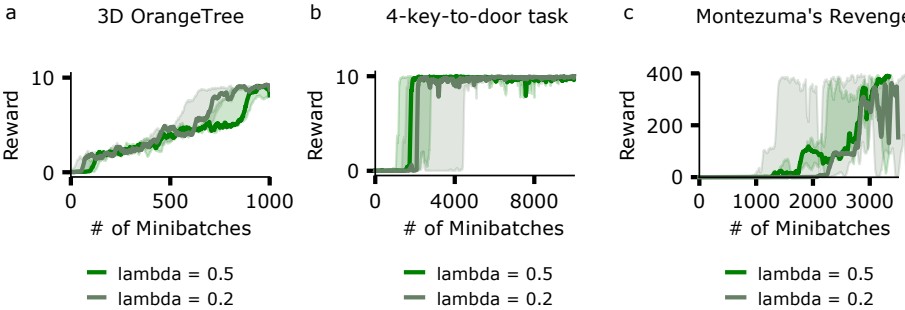

Figure A.9: ConSpec performance was relatively insensitive when $\lambda$ was changed from $0.2$ to $0.5$ on (a) the 3D OrangeTree task, (b) the 4-key task, and (c) Montezuma's Revenge. 5 seeds each condition.

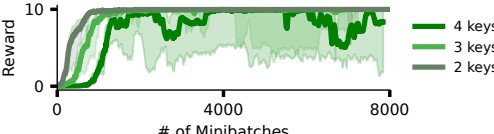

Figure A.10: Performance of ConSpec implemented on an RMA rather than PPO backbone, on the multi-key-to-door tasks. As shown, ConSpec was successfully able to rapidly learn and converge in each multi-key-to-door task, illustrating its flexibility in credit assignment, regardless the underlying RL agent.

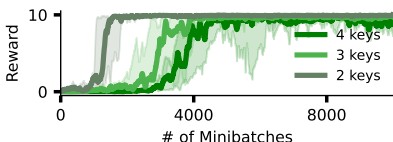

Figure A.11: Performance of ConSpec with the policy-invariant version of the intrinsic reward (equation 3) on the multi-key-to-door tasks. As shown, ConSpec was successfully able to rapidly learn each task.

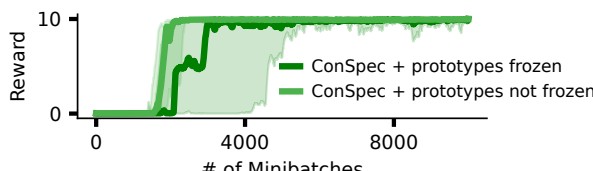

Figure A.12: Ablation of the prototype freezing in ConSpec causes no learning deficits in the multikey-to-door task.

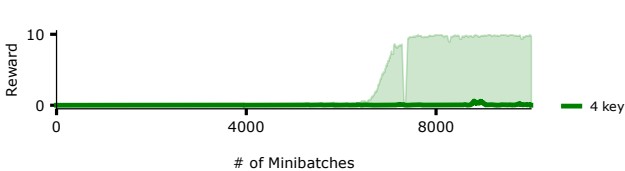

Figure A.13: ConSpec could not solve the 4-key task with a stop-gradient on the prototypes.

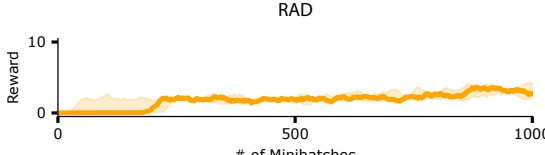

Figure A.14: Performance of RAD on the 3D OrangeTree task.

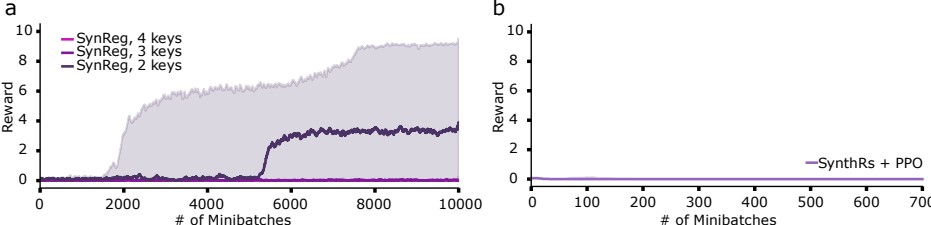

Figure A.15: Performance of SynthRs with sigmoid on (a) the multi-key-to-door and (b) OrangeTree tasks.

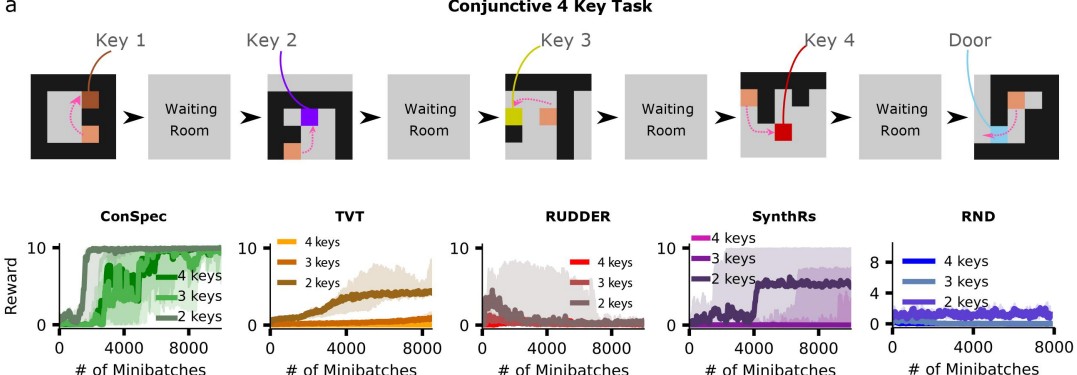

Figure A.16: ConSpec on another set of multi-key-to-door tasks with a different, conjunctive structure. Multiple keys had to be learned to be picked up, but picking up each subsequent key did not require previous keys to have been successfully picked up, unlike the paper-acceptance inspired tasks in Fig. 4. (a) Protocol for this conjunctive 4 keys task. (b) ConSpec rapidly learns these conjunctive multi-key-to-door tasks, whereas (c-f) TVT, RUDDER, SynthRs, and RND performances collapse again.

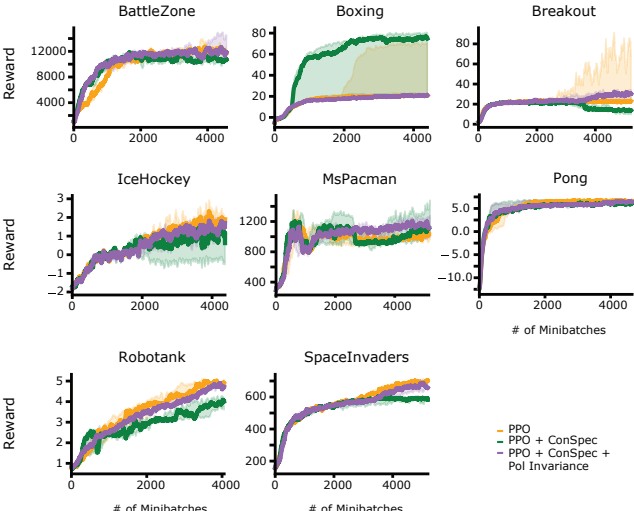

Figure A.17: PPO vs PPO+ConSpec vs PPO+ConSpec+policy invariant intrinsic reward all learn to perform similarly well on Atari Gym tasks.

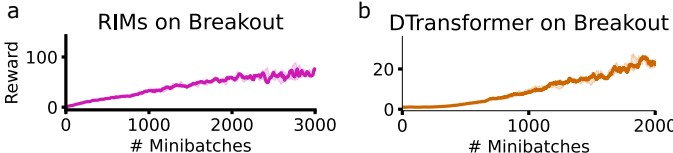

Figure A.18: Positive controls: Decision transformers and RIMs succeed in learning Atari Breakout, unlike Montezuma's Revenge in Fig. 6

