# OpenReview forum: "Contrastive Retrospection: honing in on critical steps for rapid learning and generalization in RL"
_NeurIPS.cc/2023/Conference — NeurIPS 2023 poster_

### Official Review · Reviewer_viiH · 2023-07-03

**Soundness:** 3 good
**Presentation:** 3 good
**Contribution:** 3 good
**Rating:** 6
**Confidence:** 3

**Summary:**

This paper proposes contrastive introspection (ConSpec), an algorithm for learning a set of prototypes for critical states via the contrastive loss. ConSpec works by delivering intrinsic rewards when the current states match one of the prototypes. This paper also conducted experiments in various environments.

**Strengths:**

The intuition of learning the critical states is natural and easy to follow. The experimental results in this paper look solid and promising.

**Weaknesses:**

Despite the empirical performance, the reviewer finds the ConSpec algorithm itself hard to follow.

The largest weakness is: the insufficient discussion on how the prototypes $h_i$ are learned. Hence, the reviewer cannot understand the detailed on how $h_i$ are used (see detailed in Questions).

Besides insufficient discussion on the prototypes, some minor issues are: (1) the title in the pdf (Contrastive Introspection:. ..) seems to mismatch with the one appear in openreview (ConSpec: …). (2) The font of citations appears to be confusing. E.g., from line 19-20 in the introduction, the manuscript uses (number) to address some key points, and the citation also appears as (number) – it would be nice if the citation can be changed to something that is not (number; number).


**Questions:**

Per the major weaknesses:
1. How are the prototypes $h_i$ actually learned? If the reviewer understands correctly, in line 7 of the abstract, the manuscript says “ConSpec learns a set of prototypes…”. While in Algorithm 1, it seems that the prototypes $h_i$ are given to the algorithm as inputs. Maybe the author can clarify why this inconsistency in learning the prototypes happens?
2. How are the $h_i$ learned/chosen in each experiment? The reviewer has looked into the detail of the experiments in the appendix, but cannot clearly understand how the presented experiments actually utilize the $h_i$. It would be nice that the authors can provide more details of all the $h_i$ in all the present experiments (Sec. 4.1-4.5).


**Limitations:**

See questions and weaknesses.

---

> ### Author Rebuttal · Authors · 2023-08-10
>
> We thank the Reviewer for their feedback.
>
> > *some minor issues are: (1) the title in the pdf (Contrastive Introspection:. ..) seems to mismatch with the one appear in openreview (ConSpec: …). (2) The font of citations appears to be confusing. E.g., from line 19-20 in the introduction, the manuscript uses (number)*
>
> Thank you for bringing our attention to these issues. We will address them in turn during revision; we will correct the title, and change the numbering of bullet points to be (i-iv) rather than (1-4) to minimize conflict with the numbered citations (Unfortunately, space constraints prevent us from altering the citations to an author list).
>
> **Main questions:**
> To answer the Reviewer's main question, the prototypes are initialized as random vectors, and these are provided as initial parameters, per Algorithm 1. However, these initial random vectors are updated based on the gradient of the contrastive loss - they are identical to any other parameter in this way. More specifically, each $h_i$ is a vector of parameters that are updated by the algorithm at step 8 of Algorithm 1. This is stated in section 3.1, 3.2 and 3.3.
>
> We believe the Reviewer's misunderstanding was due to our listing the prototypes as inputs in Algorithm 1. We see this ambiguity now, so thank you for raising this. But, we note that the prototypes are included in a set of parameters, and this same list includes the other parameters of the model (e.g. the synaptic weights $W$, $\theta$, and $\phi$). Thus, these prototypes are only given as inputs to the algorithm at the start in the same way that all the other randomly initialized parameters are. To clarify this point and avoid any other confusion we will separate out the parameters from the other inputs in Algorithm 1 in revision.
>
> Furthermore, we hope this resolves the Reviewer’s second question about how we choose the prototypes: we do *not* choose them. They should only be understood as parameters that are updated via gradient descent.
>
> .
>
> Conceptually, **ConSpec uses these prototypes in a very novel way, with a new contrastive loss, providing a fresh solution to both difficult problems of long-term credit assignment and generalization in RL, as we demonstrate through a variety of different task situations.**
>
> We hope that with this clarification on how the ConSpec procedure works, the reviewer will enjoy the merits of this novel algorithm and their score will be updated appropriately.

---

> > ### Author Response · Authors · 2023-08-12
> >
> > Dear Reviewer viiH, We wanted to say in advance our heartfelt thank you's for the time and effort you've put into both the reviews that have passed as well as the upcoming current discussion period. We know that you all have your own papers that you have to deal with during this busy time, and sincerely appreciate the time you've taken to spend on ours.
> >
> > We are so excited about this paper and its findings, so we are very much looking forward to the upcoming discussions with you! Please don't hesitate to ask us any questions big or small, and we are happy to provide any further clarifications.

---

> > ### Comment · Reviewer_viiH · 2023-08-12
> > **Response to the Rebuttal**
> >
> > Dear Authors,
> >
> > Thank you for your clarifications. Since all of my concerns/questions are properly addressed, the rating has been adjusted accordingly.
> >
> > Good luck!
> >
> > Reviewer viiH

---

> > > ### Author Response · Authors · 2023-08-13
> > >
> > > Dear Reviewer viiH,
> > > Thank you! We hope you enjoyed reading our work as much as we enjoyed conducting it. And we wish you good fortune in your own papers this year.

---

### Official Review · Reviewer_fj2y · 2023-07-07

**Soundness:** 4 excellent
**Presentation:** 4 excellent
**Contribution:** 3 good
**Rating:** 6
**Confidence:** 4

**Summary:**

The paper noticed that in real-world MDP, success is often contingent upon a small set of steps. While Bellman equation can theoretically do credit assignment over long-horizon, reward is hard to propagate under Bellman-based methods in practice. The authors therefore propose a novel algorithm that uses contrastive learning to identify critical states that final success relies on. The method uses a memory like system that can give agents intrinsic reward during training. The paper then evaluates the proposed method on a wide variety of domains and shows performance gain when the proposed method is added to RL algorithms.


**Strengths:**

The paper is based on an interesting and important insight about long-horizon credit assignment and reward learning. The proposed method is designed to explicitly improve long-term credit assignment and have shown empirical success in the evaluation.

The writing and figures are clear. The paper is easy to follow.

The evaluation covers a wide variety of RL tasks are benchmarked to back the claim of the paper.

**Weaknesses:**

1. The method assumes additional access to a "success" indicator at the end of episode. While this is commonly obtainable in gym environments, this doesn't fit into the general MDP setting and thus might limit when the algorithm can be applied.

2. The assumption about access to "success" seems privileged compared to baselines. I am wondering they will catch up with the performance of the proposed method when a success bonus is added.

3. The evaluation has #mini batches / # gradient steps as x-axis, unlike the environment steps in common RL benchmarks. I am wondering why this is the case. If this is necessary, I'd like to see convincing justifications.

4. The proposed method relies on a memory system, which may hurt generalization and might have problem when scaling up.

5. CURL+PPO doesn't seem to be a strong baseline to ablate in figure 3. I hope the authors could benchmark against RAD[https://arxiv.org/abs/2004.14990], a much stronger baseline in pixel space.

**Questions:**

I am wondering whether adding the intrinsic reward can degrade the performance of RL algorithms on common environments (aka, those environments where the final success does not depend on just a few critical steps). This shall be justified the experiments.

When the observation is partial, is the proposed method still reasonable?



**Limitations:**

1. The method requires privileged information about success of an episode.
2. I cannot see how the method can be applied to RL that has partial observation that requires recurrent policies.

---

> ### Author Rebuttal · Authors · 2023-08-10
>
> We thank the Reviewer for their questions and feedback.
>
> **Success/failure:**  As discussed in section A.3, we used only 2 simple definitions of success across tasks, and in each case, tie “success” to reward in a natural way. In the first definition, success is based on whether a reward is achieved at all, which works well for sparse reward environments. But, this does not work for dense rewards. Thus, the second definition defines success as simply being the top-k highest rewarded trajectories encountered so far. In other words, there is no fixed definition of success necessary beyond simply being the top level of reward achieved so far. **In this way, success is not privileged information; it is a direct function of the reward observable to the agent, and works with any MDP.**
>
> Moreover, we believe that exploiting the reward signal in this “indicator” way is a special strength of ConSpec. To illustrate, consider dense reward tasks like Atari, where we used the second definition for successes. In this case, ConSpec tries to pinpoint the differences in critical steps between the highest rewarded trajectories and the trajectories with average rewards. **As such, ConSpec uses the reward signal in a comparative and relativistic way, giving it the ability to hone in on what it can improve in each mini-batch, which is a very efficient approach.**
>
> .
>
> **Intrinsic rewards potential degradation of RL performance:** This is a good question and one that we wondered about too. In all of the tasks we have tested to-date, whether they are tasks that require long term credit assignment or not, performance is either drastically improved or unchanged with ConSpec, never degraded.
> We are confident that this pattern will hold across tasks. This is because degradation of RL performance is in general mitigated by adopting the “potential” form of intrinsic reward (Andrew Ng et al 1999) which provably does not alter the optimal policy of the original task. We have implemented this for ConSpec already, and tested it on key-to-door tasks as well as Atari Appendix Fig. A.8, A.12).
> But interestingly, even without the “potential” form of intrinsic reward  ConSpec still works well and has never led to degradation on any of the tasks we tested (Fig. 2,4,5,6,7). **We speculate that the reason why performance degradations did not occur with ConSpec is because ConSpec’s intrinsic reward is designed to be small and sparse, and so its unintended negative effects are mitigated.** Nonetheless, as noted, the use of a potential function for calculating the intrinsic reward provably deals with this problem.
>
> .
>
> **Partially observable observations:** Although partially observable environments were not a main focus of the paper, we do have several examples of partially observable environments here. The 3D OrangeTree task was one, since the pixel inputs are by no means a fully observable state. Amongst the Atari tasks, Montezuma’s Revenge and delayed AirRaid were both partially observable, and with these tasks the PPO policy was recursive, and ConSpec was able to give drastic improvements over baselines, so ConSpec was not hindered in any way by partial observability nor recurrent policies.
> Moreover, ConSpec is a plug-and-play module, and is separate from the policy itself. And ConSpec can be considered a representation learning algorithm in some ways (with a novel contrastive loss applied to RL). It discovers promising features separating successes from failures, and it is reasonable to expect it to be agnostic to architectural choices or environment characteristics.
>
> .
>
> **Plotting:** The Reviewer is correct to bring up the plotting conventions. The reason we plotted with # of gradient updates/minibatches is that the ConSpec module is updated based on complete trajectories, so we wanted a metric that would allow us to not only compare baselines, but also, compare across tasks.
>
> .
>
> **Memory system:**  - This is a great point. Scaling to pixel level tasks like Atari games and 3D navigation was not a problem, but testing other larger scale tasks was considered out of scope for the current study. Nevertheless, inspired by the Reviewer, we do have several experiments that begin trying to make ConSpec more compact. To start, the Montezuma experiments don’t fill up the success memory buffers before training starts (since successes are rare) and still works, showing that saving on memory does not hurt. Altogether, the total memory should scale with $~O(k)$ where $k$ is the number of prototypes, which should scale with the number of sparse critical steps, a manageable number.
>
> But now, we have a new experiment testing learning in ConSpec when the number of prototypes is less than required (e.g. 3 prototypes in the 4-key task) --- in order to study further memory reductions (Rebuttal Fig. 4). Even having fewer than necessary prototypes can often be enough to still solve the task (i.e. catching any critical step helps the agent)!
>
> .
>
> **RAD baseline:** We thank the Reviewer for this suggestion and have done this baseline now (Rebuttal Fig. 5). RAD was still unable to solve the 3D OrangeTree task, unlike ConSpec. To us, this elegantly demonstrates the special benefits of ConSpec’s version of contrastive learning over other representation learning algorithms applied to RL. ConSpec is not merely representation learning; it is a special application of contrastive learning tailored for long term credit assignment and generalization in RL, and our experiments show that it does this extremely well.
>
> .
>
> Altogether, this Reviewer brought up a number of interesting issues that we found very helpful. Through careful examination of these issues as well as interesting new experiments (e.g. 3-prototypes in 4-keys exp, and RAD), we demonstrate that these issues are very mitigatable. We hope the Reviewer may consider raising their score to reflect the novelty and utility of the ConSpec approach, and the strength of the varied experiments.

---

> > ### Author Response · Authors · 2023-08-12
> >
> > Dear Reviewer fj2y, We wanted to say in advance our heartfelt thank you's for the time and effort you've put into both the reviews that have passed as well as the upcoming current discussion period. We know that you all have your own papers that you have to deal with during this busy time, and sincerely appreciate the time you've taken to spend on ours.
> >
> > We are so excited about this paper and its findings, so we are very much looking forward to the upcoming discussions with you! Please don't hesitate to ask us any questions big or small, and we are happy to provide any further clarifications.

---

> > > ### Author Response · Authors · 2023-08-18
> > >
> > > Dear Reviewer fj2y, thank you for all your wonderful suggestions thus far! We wanted to ask, when convenient, if you have any further questions? We believe we have addressed and mitigated your concerns, and we are happy to address any remaining ones you may have.
> > >
> > > We know that you all have your own papers that you have to deal with during this busy time, and sincerely appreciate the time you've taken to spend on ours!

---

> > > > ### Comment · Area_Chair_oMqc · 2023-08-19
> > > >
> > > > Thank you for your rebuttal. As the reviewer has not responded yet, I will make sure to go through these points carefully before making a decision.

---

### Official Review · Reviewer_LNh7 · 2023-07-09

**Soundness:** 3 good
**Presentation:** 3 good
**Contribution:** 3 good
**Rating:** 6
**Confidence:** 4

**Summary:**

Proposes an auxiliary reward module to be used in RL algorithms, that learns features (‘prototypes’) of critical states in successful trajectories. For new observations, the method then uses cosine similarity to the learned features as a reward bonus. The method is evaluated on a unity-based env, grid-worlds, versions of gym,atari envs with delayed rewards, and Montezuma’s revenge.

**Strengths:**

1. Effective exploration bonus

The idea of learning invariant features across successes, and using these as a source of reward does seem to give better exploration performance, from the experiments. The Montezuma’s revenge experiments (Fig,6) are particularly compelling - the baselines PPO, RIMS (which also uses a set of discrete slot-based learned features ) and Decision Transformer all fail to obtain any reward. By creating an explicit division between success and failure episodes, conspec can then learn features that match states present in the successes, but not the failures, even from a very small number of successful trajectories (There might be other, simpler ways to get this effect however, see weakness #1). The ability of con spec to find important states critical for the task is also investigated by the authors in the simpler unity-based env, where they also visualize states closest to the learned prototypes.

2.   More expressive set for bottleneck states

Instead of learning an explicit set of states which are important (like sub-goals) as has been previously studied, this paper instead captures the notion of ‘critical states’ using learned prototypes. The advantage of this is it can flexibly capture a large set of very different states, all of which are critical. This is also beneficial because it enables zero-shot generalization in new environments (section 4.2)

3. Clarity, presentation

The paper is well motivated, written clearly, and the main idea for the algorithm is presented clearly.

**Weaknesses:**

1. Are the prototypes actually required?

Learning from data in successes that aren’t present in failures should lead to better performance, but the importance of doing this through learning prototype features is unclear. As a simple baseline, consider training a policy on only the successful set (using behavior cloning). Does this provide similar performance to con spec on Montezuma’s revenge? Is trying to capture a notion of ‘critical states’ required to learn better policies ? Can you run Decision Transformer where for each successful trajectory, every transition is labelled with a reward of 1, and for every failure trajectory, every transition is labelled with a reward of 0 ?

 2. Success/failure definition

The method relied crucially on the quality of the learned prototypes, which in turn depends on the success and failure datasets. It might not always be possible to divide up trajectories into 2 classes in this manner, in a lot of tasks performance keeps improving over time and a ‘successful’ trajectory at the beginning of training is very different from one from a converged policy. The authors do discuss this (appendix A.3), but the definition used in this paper for a successful trajectory is - ‘an episode that received one of the few rewards available, and a failure is defined as anything else’. For agents to keep learning and improving from data the notion of a success should necessarily change with time (eg - maximize the reward instead of just getting some reward).

3. Delayed reward envs

A good portion of the experiments are conducted on familiar gym, Atari envs but with a modification where the rewards are delayed. The significance of these experiments is unclear, since the delayed reward setting for these envs is not standard and widespread.

**Questions:**

Please address the questions in weakness #1.

**Limitations:**

Sufficiently addressed

---

> ### Author Rebuttal · Authors · 2023-08-10
>
> We thank the Reviewer for their feedback.
>
> **Are prototypes required?:**
> The reviewer raises an interesting question: could a system learn critical states implicitly, without resorting to the use of prototypes? As an example, they note that in theory one could use Decision Transformers (DT) and learn from the difference between successful and failed trajectories. Indeed, in Montezuma's Revenge, we have already fed the DT successful trajectories (i.e. trajectories that obtained reward) from an a priori set, as well as failures. In-line with the Reviewer’s recommendation we labeled the successes as 1’s and failure as 0’s. This is the result displayed in Fig. 6d and we found that performance, despite having access to success trajectories, was still zero. The reason is that these successful trajectories used for training did not come from a curated expert policy; rather, they were "spurious successes", random-policy trajectories that happened to get reward. This meant that most of the actions taken in these trajectories were actually not the correct actions, but a few were, enough to get reward. As such, behavioural cloning led the DT to learn a lot of terrible policies. Of course, behavioural cloning of curated expert policies with DTs works well.
>
> This brings up a critical but perhaps subtle point: unlike DTs without prototypes, **ConSpec is able to learn from spurious successes because the prototypes ignore all the states other than those that were critical for distinguishing success from failure. This is why ConSpec’s strategy of honing in on invariant intermediate states and ignoring all the rest of the noise is an important advance, and it is the reason why prototypes are necessary**. Put another way, the secret to the power of contrastively learned prototypes is that their limited capacity (each prototype is a single vector) is essentially a way to force the system to ignore other information that is not relevant to distinguishing successes and failures.
>
> As our experiments demonstrate, this makes ConSpec, courtesy of its prototypes, much better at learning as a beginner from scratch than DTs, which are designed to learn from experts.
>
> .
>
> **Definition of success/failure:**
> Happily, to clarify, the Reviewer’s recommendation was exactly what we had done. Altogether, we used only two simple definitions of success/failure in all of our tasks as outlined in Appendix A3. One of those definitions was “an episode that received one of the few rewards available”, as the Reviewer commented. But this pertained only to the binary reward setting -- e.g. where there is only 1 reward in the trajectory.
>
> For dense reward tasks like Atari and Mujoco, successes were defined (Section A.3 line 551-552 in the Appendix) as the top-k highest rewarded trajectories encountered so far. Failures were the median rewarded trajectories in the mini-batches. In this case, success and failure are defined relative to trajectories encountered so far, and as such, the definition of success changes over time as the agent improves - exactly covering the situation the Reviewer describes. **Hence, we agree with the Reviewer and have been doing exactly as the Reviewer recommends all along.** Given this point, we could have been more clear about the fact that the definition of success can change, so we will make this clear in a revised manuscript.
>
> Most interestingly, ConSpec can be a perfect inductive bias to learn from this changing reward signal! ConSpec is designed to pinpoint the relative differences between success and failure trajectories. And **we consider this relativistic design to reflect an important and subtle learning signal that ConSpec exploits but which other RL algorithms do not.**
>
>
>
> .
>
> **Delayed Atari:** Delayed Atari has been used in classic papers like RUDDER, and in more recent papers like InferNet (Ausin et al, 2021). More generally, delayed reward modifications of various regular environments is quite common, including in Synthetic Returns (2021) and in the episodic RL literature (e.g. Off Policy RL with delayed rewards, Han et al 2022), so we do not consider our modified Atari environments out of place.
>
> Moreover, we also apply ConSpec to the standard (unmodified) versions of Atari. Besides Montezuma’s Revenge, we also did so for 8 other Atari games (shown in appendix Fig. A.12). **The significance of our experiments on the delayed Atari is that they take a classic RL benchmark and make it more challenging vis-a-vis long-term credit assignment** (which is why standard RL algorithms such as PPO cannot solve the delayed versions but could solve most of the easier, unmodified Atari games).
>
> Ultimately, our use of delayed Atari was but one of many tasks to which we applied ConSpec, including grid world tasks, 3D pixel navigation tasks, regular Atari like Montezuma’s Revenge, and delayed continuous control tasks. These tasks differ from each other in the nature of their observations, their task features, exploration, and goal requirements. The sum total of all these tasks is to showcase the strong capability of ConSpec to robustly handle a wide variety of situations requiring long term credit assignment and/or generalization.
>
> .
>
> Altogether, this Reviewer brought up a number of issues that have helped us to highlight why ConSpec is an important contribution and to clarify our approach. We feel that we have even demonstrated that in some cases ConSpec is doing exactly as the Reviewer had called for. In light of these matters, we hope the Reviewer will raise their score to reflect the novelty and utility of the ConSpec approach, the strength of the varied experiments we provide, as well as the good ideas that we and they jointly thought of.

---

> > ### Author Response · Authors · 2023-08-12
> >
> > Dear Reviewer LNh7, We wanted to say in advance our heartfelt thank you's for the time and effort you've put into both the reviews that have passed as well as the upcoming current discussion period. We know that you all have your own papers that you have to deal with during this busy time, and sincerely appreciate the time you've taken to spend on ours.
> >
> > We are so excited about this paper and its findings, so we are very much looking forward to the upcoming discussions with you! Please don't hesitate to ask us any questions big or small, and we are happy to provide any further clarifications.

---

> > ### Comment · Reviewer_LNh7 · 2023-08-12
> > **DT comparison clarification**
> >
> > The standard manner of running Decision Transformer (DT) is to use the reward obtained in the environment, which is how I assumed the method was run in the paper. The comparison I had asked for a variant in which every transition of a successful trajectory is labelled with +1, and every transition of an unsuccessful trajectory is labelled with 0. The reason for this is equally weights all the data from the successful trajectory set. This would be different from running DT as originally presented, which uses rewards provided by the environment, and does not change them in this manner. Can the authors please clarify what version of DT they did run in their paper? If they haven't run the original version of DT (which uses environment reward), then this experiment must be run too. I think this analysis is important to see if prototypes are actually important, which is central to the argument in the paper.
> >
> > Thank you for the clarification on the relative changing definition of task success, and the use of delayed reward Atari envs in other work.

---

> > > ### Author Response · Authors · 2023-08-14
> > >
> > > Thank you Reviewer  LNh7 for the feedback. To clarify, the experiments conducted with the success trajectories followed the protocol of the original Decision Transformers paper in Chen, Lu et al, 2021. We note that in that paper, what each transition is labelled with is not its reward at the current timestep, but rather with its return-to-go $\hat{R}$ (Equation 2 of that paper here https://arxiv.org/pdf/2106.01345.pdf). In the case of sparse reward tasks like Montezuma's Revenge, this has the effect of labelling all state transitions in a success trajectory with return-to-go value, until the agent actually achieves the reward and then the rest of the transitions in that trajectory get zero. As such, our original experiments were quite close to that desired by the Reviewer. But the Reviewer is right that it does not end up labelling all the state transitions uniformly with the same label, since transitions after the agent achieves reward are labelled zero.
> > >
> > >
> > > Therefore, to conclusively address this, we are now conducting two new experiments that we hope are closer to the Reviewer's specifications (and please correct us if it is not). In the first, every transition in success trajectories has been labelled with $\hat{R}$ = the total return of that trajectory, so they are now all uniform within each success trajectory. In the second, we label every success trajectory with +1, per the Reviewer’s specification.
> > >
> > > We already have the results of the first experiment, which show that despite receiving successful demonstration trajectories, the Decision Transformer still could not learn to get nonzero reward in Montezuma's Revenge. This experiment therefore strongly supports our main hypothesis that ConSpec, courtesy of its prototypes, is much better at learning from scratch than Decision Transformers. We are waiting on the results of the second experiment and they will be ready within 72 hours.
> > >
> > > We hope this experiment is what the Reviewer was looking for. If not, please let us know and we will work hard to conduct more experiments. Thank you!

---

> > > > ### Author Response · Authors · 2023-08-17
> > > >
> > > > Dear Reviewer LNh7, in this addendum to our previous official comment, we wanted to update you on the results of our new DT experiment. We trained the Decision Transformer on the DQN replay dataset as was done in the original DT paper (https://github.com/kzl/decision-transformer/blob/master/atari/readme-atari.md) and supplemented these with 36 success trajectories that were obtained by running a random policy (as we elaborated above). We labeled every success trajectory with +1 per the Reviewer’s suggestion.  With this treatment of **equally weighting the data**, the Decision Transformer still could not learn to get nonzero reward in Montezuma's Revenge even after running the training as long as our original DT experiments in Fig. 6d.
> > > >
> > > > This was an insightful experiment that Reviewer LNh7 suggested because by contrast, ConSpec, courtesy of its prototypes' **unequal** treatment of data by honing in on a few critical states, required as few as 2 success trajectories to learn, or even started off with zero successes and discovered them spuriously on its own, and was able to rapidly learn the first 2 rewards in Montezuma’s Revenge without the help of any specialized exploration algorithms (Fig. 6a-b).
> > > >
> > > > Moreover, these experiments can be taken together with our other experiments showing that removing prototypes degraded performance (Fig. 6a yellow, Fig. A.2 yellow) to support our main hypothesis that the **secret to ConSpec's rapid learning capabilities is its ability to learn from spurious successes because the prototypes ignore the irrelevant states and learn to hone in on the few critical ones.**
> > > >
> > > > .
> > > >
> > > > Altogether, **ConSpec is a novel solution to traditional RL problems.** It is explicitly designed to pinpoint the relative differences between success and failure trajectories by the strategy of honing in on few critical states, and in doing so, reflects an important and subtle learning signal that ConSpec exploits but other RL algorithms do not. Therefore, 1) ConSpec already exploits successes v. failures in the way that the Reviewer, too, had envisioned. And 2) courtesy of the DT experiment that the Reviewer suggested, the special importance of ConSpec’s treating the experienced trajectories in an unequal manner and its benefits for learning from scratch have now been highlighted.
> > > >
> > > > We hope the Reviewer will raise their score in light of these discussions and the fruitful new experiment, and to reflect the novelty and utility of the ConSpec approach, the strength of the varied experiments we provide, as well as the good ideas that the Reviewer helped us arrive at. We stand ready to address any additional issues.

---

> > > > > ### Comment · Reviewer_LNh7 · 2023-08-18
> > > > > **Score updated**
> > > > >
> > > > > Thanks for running the new analysis, in light of this I have increased my score.

---

> > > > > > ### Author Response · Authors · 2023-08-18
> > > > > >
> > > > > > Dear Reviewer LNh7, A heartfelt thank you for the discussion! We believe that our work was very much strengthened by the constructive insights and suggestions you gave us! We hope you enjoyed reading our work as much as we enjoyed conducting it.

---

### Official Review · Reviewer_t9VM · 2023-07-27

**Soundness:** 2 fair
**Presentation:** 3 good
**Contribution:** 2 fair
**Rating:** 6
**Confidence:** 4

**Summary:**

This paper introduces ConSpec, a reinforcement learning (RL) algorithm designed to identify critical steps and improve performance in continuous control tasks. ConSpec utilizes contrastive learning to learn prototypes of critical steps and employs a contrastive loss to differentiate successful and failed experiences. It addresses the challenges of long-term credit assignment and generalization in RL tasks.

**Strengths:**

This article presents an interesting idea of learning to match key states in a task through contrastive learning. The writing of this paper is clear and Figure 1 is well-drawn, making it easy to quickly grasp the details of ConSpec. And the experimental results effectively demonstrate that the learned prototypes indeed match the key states in 3D Orange-Tree task and gridworld.

**Weaknesses:**

The cosine similarity measures the similarity between the prototype and the hidden state, both of which are learnable vectors. However, optimizing the contrastive learning loss with updates to both vectors may lead to extremely unstable training. In related literature on representation learning, it is common to use the stop gradient approach and optimize only one of the learnable vectors.


**Questions:**

- ConSpec achieved a return of only 400 in the Montezuma's Revenge task compared to RND, which reached a return of 7500 in its original paper. It appears that ConSpec is not as effective as RND in this regard. Both the multi-key room environment and Montezuma's Revenge involve similar logic, but the former is much simpler. So why does ConSpec outperform RND in Fig.4?
- It appears that ConSpec has significantly higher variance than the baseline algorithm in all tasks. What could be the reason for this? Further, prototype learning is crucial and can every seed match the key states?
- What does "softmaxed over t" mean in line 152? Why is it necessary to introduce softmax operation?
- How is success and failure defined in Atari and MuJoCo tasks? I think this is important, but the article lacks details in this aspect.

**Limitations:**

As presented in Section 5, the number of prototypes is task-specific and definations of  success and failures need human design.  Furthermore,  The proposed method introduces hyperparameters, such as the Diversity measure and the hyperparameter $\lambda$, that require careful tuning. Furthermore, the algorithm exhibits a significant variance in its actual performance.

---

> ### Author Rebuttal · Authors · 2023-08-10
>
> We thank the Reviewer for the feedback.
>
> **Contrastive instability with no stop gradient (SG):** To our knowledge, in self-supervised learning the SG is used (e.g. BYOL Grill et al 2020, or SimSiam Chen et al 2021) to prevent representational collapse (rather than learning instability), and only necessary when there are no negative samples (e.g. SimCLR has no SG). Since ConSpec uses positive and negative samples, there shouldn’t reasonably be SG related learning issues here. Nevertheless, **we did as the Reviewer suggested, stopping gradients to the prototype vectors, and found that it could not solve the 4-key task (Rebuttal Fig. 3), unlike the original ConSpec (Fig. 4c)**. This was likely since SGs cause the prototype vectors to remain in their initial random state, and so, do a poor job of anchoring the critical step recognition process.  We believe that learning both vectors is the correct strategy, and don’t find that it introduces any instability.
>
> **ConSpec v. RND:** The Reviewer is asking why ConSpec vastly outperforms RND, a classic exploration algorithm, on the multikey tasks and not on Montezuma’s Revenge. **This is a good question and reflects a deeper issue**. To us, our results indicate that there is not one but two axes of “difficulty”, and that it is incorrect to consider Montezuma’s Revenge to be more difficult in all respects. Montezuma’s Revenge is more challenging from an exploration stand-point and RND provides sophisticated exploration. On the other hand, RND performance collapses in the multikey tasks. In the  multikey tasks, the pixel difference between “success” and “failure” frames is often very small, and RND struggles to identify these differences. **On the contrary, our contrastive loss is designed to hone in on even very minute differences that distinguish success from failure (figure 4b)**, which is why ConSpec is superior to RND in the multikey tasks, **elegantly illustrating the benefits of ConSpec.**
>
> Given these two distinct strengths, we wish to highlight that in principle ConSpec could be combined with more powerful exploration algorithms, and so, one should be able to obtain the best of both worlds.
>
> **Variance:** We point out that ConSpec's performance across most tasks display **very little variance at** ***convergence***. This was true in 3D OrangeTree(Fig. 2b), the multi-key task (Fig. 4c), and for delayed Atari (Fig. 5). There does appear to be variance during training, but it seems to be the amount of time it takes for ConSpec to converge to the max reward that varies. In Montezuma's Revenge, variance likely comes from 1) stochasticity in the amount of time for the prototypes to independently hone in on different critical states, and 2) the time it takes to get spurious successes in the first place, just as the Reviewer suspected. But **to us this is benign variance as the seeds do eventually learn, and their performances vastly exceed other baseline algorithms**. Variance is still a major issue in the RL field at large (e.g. Bjorck et al ICLR 2022).
>
> **Softmaxing:** By “softmaxed over t” we mean that we are applying the softmax operation to the scores across time-steps in the trajectory. This was a design choice to sparsify the sequence of cosine scores for each trajectory. Our goal was to compare pairs of prototypes and force their peak cosine scores to be different in time. The softmax was one way to do this, but any equivalent method would do equally well. We will clarify this in a revised manuscript.
>
> **Definition of success/failure:**  In dense reward tasks like Atari and continuous control (Appendix A.3) successes are the top-k highest rewarded trajectories seen so far, while failures were the average trajectories in mini-batches. In RL tasks generally, since there usually is a reward signal, **adaptive thresholding can be used to turn any reward signal into an indicator function of success vs failure**. In this case, ConSpec tries to pinpoint the differences in critical steps between the successes from failures.
>
> To clarify, in all the experiments in the paper, we made use of only two simple definitions as described in Appendix A.3 (see our general response). We can move this information to the main manuscript if necessary.
>
> **Hyperparameters:** The Reviewer pointed out there are several hyperparameters ConSpec introduces. Luckily, for each hyperparameter, we either found a fixed value that worked across all tasks or a degree of robustness across a range of values:
> 1. Number of prototypes: we tested ConSpec with various numbers of prototypes and find that performance is robust across values (Rebuttal Fig. 1).
> 2. Diversity measure: Since this hyperparameter acts on cosine similarity scores, we know that it is necessarily bounded between 0 and 1. Moreover, since the point is simply to encourage some non-trivial diversity, we used common sense to select this value, and so, had a fixed value of 0.2 across all experiments. ConSpec learned well across all experiments, speaking to the basic robustness of this parameter choice.
> 3.  Lambda: there is a typo in equations (2) and (3) of the manuscript which we apologize for:  $\tilde{r_{kt}}$ should be equal to $\lambda \cdot R_{task} \cdot \sum...$ where $R_{task}$ is the nonzero reward per step in the task, under which $\lambda$’s range is bounded between 0 and 1. We found that performance was relatively insensitive to lambda between 0.2 and 0.5, as shown by new experiments across tasks (Rebuttal Fig. 2) with robust performance.
>
> Altogether, ConSpec performance was quite stable for these hyperparameters and did not require careful tuning per task.
>
> .
>
> In sum, this Reviewer brought up a number of issues that we believe actually highlight the robust strengths of ConSpec rather than detract from it. We hope that our responses here, including the new experiments (Rebuttal Fig. 1-5) address the original concerns enough to raise their score, and we are happy to address any insufficiencies.

---

> > ### Author Response · Authors · 2023-08-12
> >
> > Dear Reviewer t9VM,
> > We wanted to say in advance our heartfelt thank you's for the time and effort you've put into both the reviews that have passed as well as the upcoming current discussion period. We know that you all have your own papers that you have to deal with during this busy time, and sincerely appreciate the time you've taken to spend on ours.
> >
> > We are so excited about this paper and its findings, so we are very much looking forward to the upcoming discussions with you! Please don't hesitate to ask us any questions big or small, and we are happy to provide any further clarifications.

---

> > > ### Comment · Reviewer_t9VM · 2023-08-14
> > >
> > > Thank you for your thoughtful reply. I am willing to increase the score to 6. However, I do not agree that ConSpec is a robust and versatile plug-and-play method. If it is truly versatile plug-and-play, RND+ConSpec should have performed better than RND in Montezuma's Revenge, but the article neglects to include this experiment, which is not desirable. Additionally, the ConSpec algorithm consists of complex components, including dynamic definitions of success and failure, prototype learning, using learned prototypes for policy learning, and ensuring diversity of prototypes. I do not believe this is an easily deployable method in practice. However, the idea of learning key states in the task through contrastive learning, as presented in the article, is worth considering.

---

> > > > ### Author Response · Authors · 2023-08-14
> > > >
> > > > Dear Reviewer t9VM, thank you for the thoughtful discussion, and thank you for the score increase!
> > > >
> > > > We are glad that you agree that the idea of learning key states through contrastive learning is a novel idea worth considering. For this candidate solution to long term credit assignment, indeed, it brings forth new questions, due to its novelty. Those questions include how to define success, ensure diversity of prototypes, and how to combine ConSpec with exploration algorithms like RND in a principled manner. For the first 2 questions, we used simple solutions. We detail elsewhere in this rebuttal, and they seem to work well in all the diverse settings that we tried. And we think that all 3 of these are deep questions and interesting avenues of research, and plan to work them out in principled manners in future work, particularly the question of combining ConSpec with other exploration algorithms.
> > > >
> > > >  At the heart of it, we believe that ConSpec is a promising solution because the contrast between success and failure is a special, exploitable signal and ConSpec is overtly designed to hone in on this.
> > > >
> > > > This first ConSpec paper, for us, is a shot across the bow. And we are excited about the line of work that will come out of this because we believe that in the end, ConSpec will prove itself.
> > > >
> > > > We hope you enjoyed reading our work as much as we enjoyed conducting it! Thank you again for taking the time for our paper, and we wish you good fortune in your own papers this year.

---

### Author Rebuttal · Authors · 2023-08-09

**Message to all reviewers:**
Thank you for the comments, questions, and suggestions. In this global response we will address those issues highlighted by multiple reviewers. We will respond to individual reviewer comments in the individual responses.

First, we want to summarize and clarify the central, novel contribution of our work. Whereas most contemporary RL algorithms try to model or fit values, returns, or transitions for all the states in the environment, the present manuscript investigates an alternative learning strategy: that of retrospectively honing in on a few critical steps, courtesy of a novel contrastive loss. ConSpec tests the hypothesis that **the ability to recognize  critical steps (and ignore all other states) allows rapid long-term credit assignment and robust generalization.** In support of this hypothesis, we have demonstrated strong performances by ConSpec in a wide variety of different task environments, discrete and continuous tasks, gridworlds, 2D Atari including Montezuma’s Revenge, and 3D navigation. We managed to isolate particular weaknesses of other algorithms for long-term credit assignment including RUDDER, Synthetic Returns, and Temporal Value Transport (TVT) in situations with multiple contingencies, which are common in real life, and which ConSpec overcomes easily thanks to its retrospective credit assignment. Moreover, we even managed to **isolate particular weaknesses** of other RL strategies like RND exploration, and Decision Transformers that ConSpec can overcome, demonstrating the uniqueness of ConSpec as well as its potential complementarity to these other algorithms.

Thus, our results show that retrospectively honing in on critical steps is a potentially powerful thing to do especially for agents learning from scratch (to learn the world a bit at a time).

**New experiments**: In the individual Reviewer questions below, the Reviewers brought up a variety of interesting issues that have helped us to think more deeply about this approach. Motivated by these points, we conducted a number of small experiments. We found that all the worries that the Reviewers brought up are easily mitigated by ConSpec. **Our new experiment figures are in the attached PDF, and include:**

- Demonstrating robustness to hyperparameter choices
- A version of ConSpec with stop-gradient on prototypes
- More compact ConSpec with fewer than necessary prototypes
- A new baseline RAD (stronger than CURL)

**Success v. Failure**: Several Reviewers brought up issues related to the definition of successes (Reviewers t9VM, LNh7, fj2y), so we will give a summary here (and we have also answered their individualized questions below). As discussed in Appendix section A.3, we used only 2 simple definitions of success across all of our tasks, and in each case, we tied “success” to reward in a natural way. In the first definition, success is based on whether a reward is achieved at all, which works well for sparse reward environments, but this does not work for dense rewards. Thus, the second definition defines success as the top-k highest rewarded trajectories encountered so far, which means that the trajectories defined as successful are updated as an agent improves. As such, ConSpec uses the reward signal in a comparative and relativistic way, giving it the ability to hone in on key steps associated with the best performance achieved so far, a very efficient approach. This is a special strength of ConSpec.

In light of our findings, we believe that ConSpec is a **robust and versatile plug-and-play** module that can improve credit assignment and generalization in RL. Moreover, it is a **novel solution to traditional RL problems**. Most notably, it is a novel application of **contrastive** learning in RL, distinct from other contemporary attempts, and it takes a **retrospective** strategy to credit assignment, discovering prototypes of a few critical steps that have the capacity to both help assign credit over long time horizons and generalize, as we show with strong performance in widely varied experimental settings.

All in all, we believe that ConSpec provides a novel, well-validated approach to improving difficult credit assignment in RL, and would benefit the NeurIPS community in this venue. We hope that the reviewers agree, particularly given our new experiments, and see fit to raise their scores.

---

### Author Response · Authors · 2023-08-21
**Final message from the Authors to the Reviewers:**

We wish to express our sincere thanks to all Reviewers for the positive feedback on our work. We believe we were truly lucky to have had Reviewers that gave insightful and constructive ideas and helped strengthen our study, and will incorporate all of the reviewers' suggestions in the final revision.

Thank you again for your consideration here!

---

### Decision · Program_Chairs · 2023-09-21

**Decision:**

Accept (poster)

**Comment:**

The paper provides a unique and interesting approach to representation learning in RL. I went through the reviews and paper. I agree with the reviewers that the paper is a strong submission: Good and clear writing, interesting and novel algorithmic approach, well-designed experiments (thanks to the reviewers who encouraged authors to run a few more ablations/baselines!).